# Extending Contextual Self-Modulation: Meta-Learning Across Modalities, Task Dimensionalities, and Data Regimes

## Abstract

Contextual Self-Modulation (CSM) is a potent regularization mechanism for Neural Context Flows (NCFs) which demonstrates powerful meta-learning on physical systems. However, CSM has limitations in its applicability across different modalities and in high-data regimes. In this work, we introduce two extensions: $i$CSM, which expands CSM to infinite-dimensional tasks, and StochasticNCF, which improves scalability. These extensions are demonstrated through comprehensive experimentation on a range of tasks, including dynamical systems with parameter variations, computer vision challenges, and curve fitting problems. $i$CSM embeds the contexts into an infinite-dimensional function space, as opposed to CSM which uses finite-dimensional context vectors. StochasticNCF enables the application of both CSM and $i$CSM to high-data scenarios by providing a low-cost approximation of meta-gradient updates through a sampled set of nearest environments. Additionally, we incorporate higher-order Taylor expansions via Taylor-Mode automatic differentiation, revealing that higher-order approximations do not necessarily enhance generalization. Finally, we demonstrate how CSM can be integrated into other meta-learning frameworks with FlashCAVIA, a computationally efficient extension of the CAVIA meta-learning framework (Zintgraf et al. 2019). FlashCAVIA outperforms its predecessor across various benchmarks and reinforces the utility of bi-level optimization techniques. Together, these contributions reaffirm the powerful benefits of CSM, and suggest that its spectrum of addressable meta-learning and out-of-distribution tasks is limited to physical systems. Our open-sourced library, designed for flexible integration of self-modulation into contextual meta-learning workflows, is available at `AnonymousGitHubRepo`.

## 1 Introduction

Meta-learning has emerged as a powerful paradigm in machine learning, addressing the limitations of conventional approaches that train a single algorithm for a specific task. This innovative technique aims to develop models capable of rapid adaptation to novel but related tasks with minimal data, a process often referred to as "learning to learn" (Wang et al., 2021). By leveraging common information across multiple training environments (or meta-knowledge), meta-learning algorithms can efficiently adapt to new scenarios without starting from scratch (Hospedales et al., 2021). The success of meta-learning has been demonstrated in various domains, including dynamical system reconstruction (Norcliffe et al., 2021), program induction (Devlin et al., 2017), out-of-distribution (OoD) generalization (Yao et al., 2021), and continual learning (Hurtado et al., 2021).

Recent advancements in meta-learning have focused on reducing the number of adaptable parameters to a smaller subset known as a "**context**", which encodes environment-specific information. This approach, termed "contextual meta-learning", has shown superior performance in terms of speed, memory efficiency, and accuracy compared to alternative methods (Zintgraf et al., 2019; Garnelo et al., 2018; Gordon et al., 2019; Nzoyem et al., 2024). Given the growing popularity of contextual meta-learning, a comprehensive comparison of some of its methods is essential to guide its users. In this paper, we identify three critical axes of comparison—task modality, task dimensionality, and data regime—necessary for evaluating the general competitiveness and applicability of contextual meta-learning approaches.

**(R1 - Task Modality)**   Contextual meta-learning methods have demonstrated remarkable success across various data modalities, including images, meshes, audio, and functas (Dupont et al., 2022). However, recent observations have revealed limitations in their performance on time series data from physical systems (Kirchmeyer et al., 2022). For instance, CAVIA (Zintgraf et al., 2019) with its bi-level optimization algorithm has been reported to overfit when learning the underlying parameter-dependence of dynamical systems (Nzoyem et al., 2024). Conversely, the Neural Context Flow (NCF) (Nzoyem et al., 2024), which has recently achieved state-of-the-art results on several dynamical systems benchmarks, remains untested in decision-making scenarios and other domains where established contextual meta-learning methods excel.

**(R2 - Task Dimensionality)**   The ability to adapt to infinite-dimensional changes, rather than fixed-size vector embeddings, is increasingly demanded of contextual meta-learning. Such demands are pressing in physical systems learning (Yin et al., 2021; Mishra et al., 2017; Nzoyem et al., 2024), where a common challenge is generalizing to parameter changes in the underlying dynamical system, such as the time-invariant gravity $g$ of a swinging pendulum. While several recent works have successfully modelled such time-invariant parameter changes through contextual meta-learning (Liu et al.; Day et al., 2021), many approaches have overlooked cases where the changing parameter is itself a function of time, such as the forcing term $f(\cdot)$ of a pendulum.

**(R3 - Data Regime)**   While meta-learning is designed to require limited data during the meta-testing stage (Hospedales et al., 2021), the optimal amount of data needed for effective meta-training remains an open question. This issue is particularly evident in image completion tasks, where Conditional Neural Processes (CNPs) perform well in low-data regimes but struggle to reconstruct known pixels in high-data scenarios (Gordon et al., 2019). Understanding the necessary adjustments for meta-learning in both low and high-data regimes is crucial, especially given the potential contradictions between neural scaling laws (Kaplan et al., 2020; Hoffmann et al., 2022) and monotone learning in some scenarios (Bousquet et al., 2022).

Contextual Self-Modulation (CSM) is a recently proposed regularization mechanism for smooth physical systems (Nzoyem et al., 2024) that shows promise in addressing the aforementioned requirements. Importantly, it can be combined with other contextual meta-learning techniques. This work examines several families of contextual meta-learning approaches through the lens of CSM. In the following sections, we present a common problem setting followed by a brief summary of the methods involved.

## 1.1   Problem Setting

We consider two distributions $p_{\text{tr}}(\mathcal{E})$ and $p_{\text{te}}(\mathcal{E})$ over (meta-)training and (meta-)testing environments (or tasks), respectively. The former is used to train the model to learn how to adapt to given tasks, while the latter evaluates its ability to quickly adapt to previously unseen but related environments with limited data. Adaptation is qualified as In-Domain (InD) when $p_{\text{tr}} = p_{\text{te}}$, and Out-of-Distribution (OoD) otherwise. From either distribution, we assume a maximum of $N$ distinct environments can be sampled.

In the typical regression setting, our goal is to learn a mapping $f : x \mapsto y$, where $x \in \mathcal{X}$ is a datapoint and $y \in \mathcal{Y}$ is its corresponding label. In this work, all environments share the same $\mathcal{X}, \mathcal{Y}$, and loss function $\mathcal{L}^e$. Each environment $e$ is defined by a distribution $q^e(x, y)$ over labelled datapoints. We sample two datasets from $q^e$: the **training** (or support) set $\mathcal{D}_{\text{tr}}^e = \{(x^e, y^e)^m\}_{m=1}^{M_{\text{tr}}^e}$ and the **test** (or query) set $\mathcal{D}_{\text{te}}^e = \{(x^e, y^e)^m\}_{m=1}^{M_{\text{te}}^e}$, with $M_{\text{tr}}^e$ and $M_{\text{te}}^e$ representing the number of support and query datapoints, respectively. Adaptation to environment $e$ is performed using the former set, while performance evaluation is done on the later.

In contextual meta-learning, we train a model $f : x^e, \theta, \xi^e \mapsto \hat{y}^e$, where $\theta \in \mathbb{R}^{d_\theta}$ are $d_\theta$-dimensional learnable parameters shared across all environments (e.g., neural network weights and biases), and $\xi^e \in \Xi$ are task-specific contextual parameters that modulate the behavior of $\theta$. Although we generally define $\Xi$ as a subset of $\mathbb{R}^{d_\xi}$, it is important to note that this definition may obscure additional considerations (cf. Section 2).

Dynamical system reconstruction (Göring et al., 2024; Kramer et al., 2021) can be viewed as a supervised learning problem. In this case, the predictor maps from a bounded set $\mathcal{A}$ to its tangent

bundle $T\mathcal{A}$, forming an evolution term to define a differential equation over a time interval $[0, T]$:

$$\frac{\mathrm{d}x_t^e}{\mathrm{d}t} = f(x_t^e, \theta, \xi^e). \tag{1}$$

In this work, trajectories $x_t^e$ are computed using differentiable numerical integrators. During data generation, the initial condition, $x_{t_0}^e$, which determines the subsequent trajectory $x_t^e$, is sampled from a known distribution for both meta-training and meta-testing. Meanwhile, the underlying physical parameters that define the vector field are sampled from either $p_{\mathrm{tr}}$ or $p_{\mathrm{te}}$, as previously described.

## 1.2 RELATED WORK

Contextual meta-learning methods aim to modulate the behavior of a main network with latent information. Our work broadly encompasses three families of contextual meta-learning methods: the Neural Processes family, Gradient-Based Meta-Learning (GBML), and Neural Context Flows (NCFs). The Neural Processes family, represented here by the CNP (Garnelo et al., 2018), leverages an encoder $g_\phi$ with input-output pairs to construct its contextual representation, which is subsequently processed by a decoder $f_\theta$. GBML approaches employ a **bi-level** optimization scheme during meta-training and, unlike Neural Processes, require a form of gradient descent even during adaptation. In this work, we investigate the GBML family through the lens of CAVIA (Zintgraf et al., 2019). The NCF (Nzoyem et al., 2024), discards the bi-level optimization scheme for a more computationally efficient proximal **alternating** scheme. Table 1 presents several differences and similarities between representatives of these families. Additional algorithmic details on each method, along with other bodies of work relevant to R1-R3, are presented in Appendix A.

Table 1: Comparison of contextual meta-learning approaches for generalization on a typical regression task. The Memory column accounts for the size of the context-fitting component in the framework during training. We expect better contexts when this component increases in size. The Computation column indicates the cost of making $m$ predictions based on $n$ few-shot adaptation points (used to generate context vectors). The notation $|X|$ denotes the number of elements in $X$ (Park et al., 2023).

| METHOD | PARAMETERS | ADAPTATION RULE | MEMORY | COMPUTATION |
|--------|-----------|-----------------|--------|-------------|
| **CNP** | $f_\theta, g_\phi$ | $\xi^e = g_\phi(\mathcal{D}_{\mathrm{tr}}^e)$ | $O(|\phi|)$ | $O(n + m)$ |
| **CAVIA** | $f_{\{\theta, \xi^e\}}$ | Inner gradient updates with $H$ steps | $O(|\theta| \cdot H)$ | $O((n + m) \cdot H)$ |
| **NCF** | $f_{\{\theta, \xi^e\}}$ | Gradient descent with $H$ steps | $O(1)$ | $O((n + m) \cdot H)$ |

## 1.3 CONTRIBUTIONS

Our primary contribution is the systematic examination of Contextual Self-Modulation (CSM) in various task and data settings, exploring strategies to mitigate issues that arise, and investigating its potential to enhance established meta-learning techniques. Specifically, we contribute the following four points:

**(1)** We extend the CSM regularization mechanism to infinite dimensions ($i$CSM) and arbitrarily high Taylor orders, which we successfully apply to complex dynamical systems with functional parameter changes and beyond. Notably, we show that $i$CSM performs consistently well, even on finite-dimensional problems, thus generalizing CSM.

**(2)** We propose StochasticNCF, a stochastic generalization of Neural Context Flows (NCF) (Nzoyem et al., 2024) designed to handle the computational demands of high-data regimes like curve-fitting and image completion. The randomness it introduces is compatible with both CSM and $i$CSM, and enables robust comparison of NCF against gradient-based meta-learning which has such a randomness feature built-in.

**(3)** We present FlashCAVIA, a powerful update of CAVIA (Zintgraf et al., 2019). This implementation demonstrates the compatibility of CSM with bi-level adaptation rules. Together, they enable more efficient use of numerous inner updates, leading to more expressive models.

**(4)** We develop a software library that facilitates the application of all aforementioned strategies and the combination thereof. The code, along with several examples, is made open-source and available at `AnonymousGitHubRepo`.

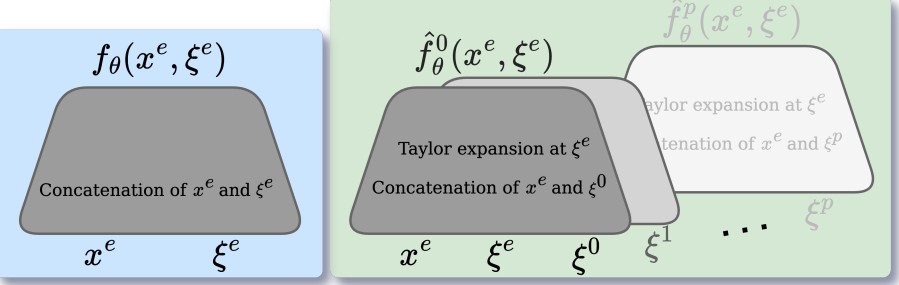

Figure 1: **(Left)** Illustration of the classical setting of concatenation-based conditioning (Dumoulin et al., 2018; Zintgraf et al., 2019) for predictions in the environment $e$. **(Right)** Illustration of Contextual Self-Modulation (CSM) where, in addition to concatenation, Taylor expansion as defined in Eq. (2) is performed at $\xi^e$ using neighbouring contexts $\xi^j$, with $j = 0, \ldots, p$ (Nzoyem et al., 2024). In our $i$CSM setting, $x^e$ is concatenated to $\xi^j(\cdot)$ rather than $\xi^j$ itself.

## 2 METHODS

### 2.1 CONTEXTUAL SELF-MODULATION

Contextual Self-Modulation (CSM) is an innovative technique that enhances a function's adaptability across various contexts by leveraging its inherent smoothness with respect to contextual parameters (Nzoyem et al., 2024). Consider a function $f_\theta : \mathcal{X} \times \Xi \to \mathcal{Y}$, where $\mathcal{X}$ is the input space, $\Xi$ is the context space, and $\mathcal{Y}$ is the output space. CSM generates a set of candidate approximations for any target context $\xi^e \in \Xi$, using a **pool** of known nearest[1] contexts $\mathrm{P} = \{\xi^1, \ldots, \xi^p\} \subset \Xi$.

The core of CSM lies in the generation of candidate approximations $\{\hat{f}_\theta^j\}_{j=1}^p$ using the $k$-th order Taylor expansion of $f_\theta$ around each $\xi^j \in \mathrm{P}$

$$\hat{f}_\theta^j(x^e, \xi^e) = \sum_{n=0}^{k} \frac{1}{n!} \nabla_\xi^n f_\theta(x^e, \xi^j) \otimes (\xi^e - \xi^j)^n, \tag{2}$$

where $\nabla_\xi^n f$ denotes the $n$-th order tensor derivative of $f$ with respect to $\xi$, and $(\xi^e - \xi_j)^n$ is the $n$-fold tensor product. The discrepancy between these candidate approximations and the ground truth label is then minimized

$$\min_{\theta, \xi^e} \mathcal{L}^e(\theta, \xi^e, \mathcal{D}^e) := \min_{\theta, \xi^e} \mathbb{E}_{(x^e, y^e)} \left[ \frac{1}{p} \sum_{j=1}^{p} \ell(\hat{f}_\theta^j(x^e, \xi^e), y^e) \right], \tag{3}$$

where $\ell$ is an appropriate loss function. This formulation is illustrated in Fig. 1; it allows the predictor to seamlessly interpolate and extrapolate across the context space, drawing insights from known contexts to improve performance in novel situations.

**Finite dimensions.** The original CSM methodology (Nzoyem et al., 2024) proposes $\Xi \subset \mathbb{R}^{d_\xi}$. In that approach, the context is pre-processed by an optional context network, then concatenated to (a pre-processed) $x^e$ before feeding into a main neural network.

**Infinite dimensions.** We extend CSM to infinite-dimensional variations by letting $\Xi$ be a function space, naming this approach $i$CSM (see Fig. 1). This extension allows our model $f_\theta$ to adapt to a broader range of changes. In our implementation of $i$CSM, we chose $\Xi$ as the space of multi-layer perceptrons, whose weights are flattened into a 1-dimensional tensor to perform Taylor expansion. The input $x^e$ is no longer concatenated to this 1-dimensional tensor $\xi^e$, but rather to its output $\xi^e(\cdot)$ – the context is viewed as a function of another variable, typically time $t$ when learning dynamical systems. To ensure stability of our predictive mapping, we initialize the weights of these neural networks at 0 since Taylor expansions are local approximations and perform better when $\xi^e$ is close to $\xi^j$. Random initialization of neural networks does not guarantee such proximity, and leads to divergence early in the training process.

---

[1]Proximity is calculated in $L^1$ norm over the space $\Xi$, and all $p$ contexts in the pool must be distinct.

**Higher-order Taylor expansions.**    With Proposition 3.1, Nzoyem et al. (2024) propose a technique to efficiently calculate the second-order Taylor expansion without materializing Hessians. However, their approach is limited to second order, and a naive calculation of higher-order derivatives would incur an exorbitant cost of $O(\exp(k))$ due to redundant recomputations (Bettencourt et al., 2019). In this work, we leverage Taylor-mode automatic differentiation (AD) to compute these derivatives at a much lower cost. Specifically, we adapt the `jet` API (Bradbury et al., 2018) to derive $\hat{f}_\theta^j$ in Eq. (2). While `jet` requires both a primal value and a series of coefficients to calculate the derivatives of $f = g \circ h$, our setting is not concerned with function composition. As such, we set our primal value to $\xi^j$ and the demanded series to $[\xi^e - \xi^j, \mathbf{0}, \dots, \mathbf{0}]$. The output sequence is then multiplied by the appropriate inverse factorials and summed to produce the Taylor approximation. For the remainder of this work, any Taylor expansion of order $k > 2$ leverages the `jet` API.

**Benefits of self-modulation.**    A key advantage of both CSM and $i$CSM is uncertainty quantification (Nzoyem et al., 2024). Upon training, we obtain multiple contexts in close proximity, allowing us to produce an ensemble of candidate predictions in order to ascertain the variance inherent in the task distribution through the learned contexts (see for example Figs. 11 and 12).

**Perceived limitations.**    We acknowledge that not all functions can be approximated near $\xi^0 \in \Xi$ by Taylor expansion, as such guarantees are only valid for analytic functions within a prescribed radius of convergence $R > 0$ (Cartan, 1995). In Appendix B.2, we address the question of whether our power series in Eq. (2), modulated through CSM, can recover discrepancies when the underlying parameters are farther apart than $R$.

**CSM for NCF.**    In the framework of Neural Context Flows (NCF) (Nzoyem et al., 2024), CSM refers specifically to the Taylor expansion process used to produce candidate predictions. It does not encompass other concepts, such as the 3-networks architecture[2], penalization, or the alternating optimization scheme. In fact, we introduce a joint-optimization variant termed NCF* in Section 3.4, demonstrating improved performance over NCF while keeping the same CSM mechanism.

## 2.2 STOCHASTICNCF

StochasticNCF introduces a stochastic element to the Neural Context Flows framework, allowing for training on **excessively large** datasets. We denote by $\mathcal{L}$ the mean loss across environments:

$$\mathcal{L}(\theta, \xi, \mathcal{D}) = \frac{1}{N} \sum_{e=1}^{N} \mathcal{L}^e(\theta, \xi^e, \mathcal{D}^e), \tag{4}$$

where $\xi := \{\xi^e\}_{e=1}^N$ is the union of all learning contexts. At each iteration, we approximate $\nabla_\theta \mathcal{L}$ and $\nabla_\xi \mathcal{L}$ using the gradients of only a few indices when evaluating $\mathcal{L}(\theta, \xi, \mathcal{D})$. Those indices are collectively grouped in $B \subset \{1, 2, ..., N\}$ with cardinality $|B|$ set as a hyperparameter. The minibatch $B$ is selected using a "nearest-first" approach[3]: we randomly sample the first element $e*$ from $\mathcal{U}\{1, N\}$, then select the remaining $|B| - 1$ indices based on their $L^1$-proximity to $e*$ in the context space. We use the unbiased SGD estimator for gradient estimation

$$\tilde{\nabla}_\theta \mathcal{L}(\theta, \xi, \mathcal{D}) = \frac{1}{|B|} \sum_{e \in B} \nabla_\theta \mathcal{L}^e(\theta, \xi^e, \mathcal{D}^e), \tag{5}$$

while remarking that this approach is compatible with other biased and unbiased gradient estimators, e.g. SAGA, SARAH (Driggs et al., 2021).

This approach not only accelerates training by enabling more **computationally efficient** steps, but also prevents forced information sharing across unrelated or distant environments. It augments the ability of the model to automatically discriminate clusters of environments. In the rest of this paper, we use NCF to refer to both StochsticNCF and deterministic NCF, with the understanding that deterministic NCF is equivalent to StochasticNCF with the maximum number of loss contributors, i.e. $|B| = N$.

---

[2]When evaluating the $i$CSM implementation, we found the "context network" from (Nzoyem et al., 2024) to provide no expressivity benefits, which motivated its removal.

[3]Note that the "nearest-first" approach performed at this stage is in addition to the one performed in the context pool P.

## 2.3 FLASHCAVIA

To demonstrate the applicability of CSM within other meta-learning frameworks, we redesigned and implemented CAVIA (Zintgraf et al., 2019) from scratch, resulting in FlashCAVIA. This derivative framework is designed for greater performance compared to the original CAVIA implementation, while incorporating CSM at the model level.

Some key improvements in FlashCAVIA include:

(i) **Parallelization:** We parallelize the sequential task loop (line 6 in Algorithm 1) to process all environments simultaneously, allowing for fairer comparison with the three-way parallelized NCF (Nzoyem et al., 2024).

(ii) **Efficient inner updates:** For each inner gradient loop, we use the prefix sum primitive `scan` (Bradbury et al., 2018) to perform longer and more efficient inner updates. This follows a recent trend with efficient hardware-aware implementations of state-space models (Gu & Dao, 2023).

(iii) **Custom optimizer:** We leverage a custom optimizer to steer the inner gradient updates, which is particularly important when performing a large number of inner gradient updates (e.g., $H = 100$ in Section 3.1).

Although it requires no changes at the model architecture level, our main contribution to FlashCAVIA is **the integration of CSM**. The forward-mode Taylor expansion step may require a bespoke reverse-mode automatic differentiation depending on the AD mode used at higher levels of the optimization process, and whether custom adjoint rules are involved.

These improvements in FlashCAVIA not only enhance its performance but also provide a versatile platform for comparing different meta-learning approaches. They also allow the exploration of the benefits of CSM across various frameworks.

## 3 EXPERIMENTAL SETUP & RESULTS

This section presents a comprehensive analysis of the CSM mechanism's performance across four distinct problem domains: curve-fitting, optimal control, dynamical system reconstruction, and image completion. We examine how requirements R1 to R3 are met in these varied settings. For each experiment, we describe the dataset and analyze the results and their implications. The training hyperparamters are detailed in Appendix B.

### 3.1 SINE REGRESSION

The sine regression experiment, a standard benchmark in meta-learning (Finn et al., 2017), serves as our initial test for the CSM mechanism. This curve-fitting regression problem allows us to assess its benefits in NCF. This experiment also evaluates the impact of $H \in \{1, 5, 100\}$ gradient update iterations on the CAVIA (Zintgraf et al., 2019) and MAML (Finn et al., 2017) baselines, and in our FlashCAVIA implementation. We generate input-output pairs based on the generalized sinusoid $y = A \sin(x - \alpha)$, where amplitude $A \in [0.1, 5.0]$ and phase $\alpha \in [0, \pi]$ are sampled from uniform distributions. The same distributions are used for both meta-training and adaptation. We explore low-, medium-, and high-data regimes with $N = 250$, $N = 1000$, and $N = 12500$ environments, respectively.

The results of our large-scale sine regression experiment comparing MAML, CAVIA, FlashCAVIA, and NCF across various data regimes, are presented in Table 2. MAML and the original CAVIA implementation demonstrate performance consistent with (Zintgraf et al., 2019), showing minimal improvement when scaling inner gradient updates to 100. In contrast, FlashCAVIA exhibits significant benefits from 100 inner updates, surpassing NCF in high-data regimes ($N = 125000$). Notably, all GBML methods display poorer results with larger $|B|$, suggesting potential issues with meta-gradient descent directions. Conversely, NCF performance improves with increased tasks per meta-update. An important observation is the larger error bars for NCF compared to GBML baselines, indicating greater variability in environment resolution.

Table 2: Results for the sine curve-fitting experiment with varying numbers of environments $N$, for both small and large number of tasks per meta-update $|B|$. We report the mean MSE across evaluation environments with one standard deviation. The number of inner gradient updates $H$ is indicated following the method's name. FlashCAVIA-100 consistently outperforms others across all columns (all shaded in grey). This experiment includes further runs which were averaged across Taylor orders $k$ and context size $d_\xi$, and we direct the reader to Fig. 6 for additional details including those.

| $|B| = 25$ | $N = 250$ | | $N = 1000$ | | $N = 12500$ | |
|---|---|---|---|---|---|---|
| | TRAIN | ADAPT | TRAIN | ADAPT | TRAIN | ADAPT |
| **MAML-1** | $2.02 \pm 1.01$ | $1.98 \pm 1.07$ | $2.36 \pm 1.35$ | $1.90 \pm 0.97$ | $2.05 \pm 1.15$ | $1.97 \pm 1.16$ |
| **CAVIA-1** | $0.53 \pm 0.15$ | $0.49 \pm 0.08$ | $0.47 \pm 0.10$ | $0.43 \pm 0.06$ | $0.49 \pm 0.14$ | $0.50 \pm 0.12$ |
| **FLASHCAVIA-1** | $1.28 \pm 0.47$ | $1.52 \pm 0.41$ | $1.37 \pm 0.40$ | $1.53 \pm 0.38$ | $0.98 \pm 0.32$ | $1.09 \pm 0.33$ |
| **MAML-5** | $1.76 \pm 0.84$ | $1.78 \pm 0.82$ | $1.82 \pm 0.91$ | $1.79 \pm 0.92$ | $2.07 \pm 1.07$ | $1.77 \pm 0.79$ |
| **CAVIA-5** | $0.41 \pm 0.16$ | $0.42 \pm 0.16$ | $0.42 \pm 0.23$ | $0.41 \pm 0.22$ | $0.42 \pm 0.21$ | $0.40 \pm 0.21$ |
| **FLASHCAVIA-5** | $0.19 \pm 0.05$ | $0.27 \pm 0.06$ | $0.23 \pm 0.10$ | $0.29 \pm 0.11$ | $0.12 \pm 0.06$ | $0.17 \pm 0.09$ |
| **MAML-100** | $3.76 \pm 0.13$ | $3.84 \pm 0.30$ | $3.73 \pm 0.16$ | $3.73 \pm 0.45$ | $4.13 \pm 0.05$ | $3.86 \pm 0.08$ |
| **CAVIA-100** | $1.76 \pm 0.47$ | $1.99 \pm 0.65$ | $1.79 \pm 0.74$ | $1.61 \pm 0.78$ | $2.51 \pm 1.23$ | $2.36 \pm 1.16$ |
| **FLASHCAVIA-100** | $0.0012 \pm 0.0008$ | $0.0040 \pm 0.0026$ | $0.0245 \pm 0.0490$ | $0.0349 \pm 0.0607$ | $0.0004 \pm 0.0004$ | $0.0013 \pm 0.0010$ |
| **NCF** | $0.022 \pm 0.035$ | $0.460 \pm 0.342$ | $0.045 \pm 0.035$ | $0.148 \pm 0.151$ | $0.222 \pm 0.055$ | $0.047 \pm 0.036$ |

| $|B| = 250$ | $N = 250$ | | $N = 1000$ | | $N = 12500$ | |
|---|---|---|---|---|---|---|
| | TRAIN | ADAPT | TRAIN | ADAPT | TRAIN | ADAPT |
| **MAML-1** | $3.95 \pm 0.68$ | $3.91 \pm 0.62$ | $3.95 \pm 0.68$ | $3.91 \pm 0.62$ | $3.71 \pm 0.71$ | $4.50 \pm 0.72$ |
| **CAVIA-1** | $3.40 \pm 0.60$ | $3.39 \pm 0.57$ | $3.42 \pm 0.61$ | $3.40 \pm 0.57$ | $3.34 \pm 0.62$ | $4.11 \pm 0.67$ |
| **FLASHCAVIA-1** | $1.22 \pm 0.48$ | $1.46 \pm 0.45$ | $1.33 \pm 0.40$ | $1.49 \pm 0.37$ | $1.29 \pm 0.44$ | $1.41 \pm 0.42$ |
| **MAML-5** | $1.76 \pm 0.84$ | $1.78 \pm 0.82$ | $4.12 \pm 0.73$ | $4.04 \pm 0.65$ | $3.75 \pm 0.73$ | $4.53 \pm 0.73$ |
| **CAVIA-5** | $0.41 \pm 0.16$ | $0.42 \pm 0.16$ | $3.95 \pm 0.69$ | $3.90 \pm 0.62$ | $3.64 \pm 0.70$ | $4.42 \pm 0.71$ |
| **FLASHCAVIA-5** | $0.29 \pm 0.29$ | $0.36 \pm 0.30$ | $0.26 \pm 0.08$ | $0.33 \pm 0.07$ | $0.25 \pm 0.06$ | $0.31 \pm 0.06$ |
| **MAML-100** | $3.76 \pm 0.13$ | $3.84 \pm 0.30$ | $4.51 \pm 0.81$ | $4.61 \pm 0.78$ | $4.53 \pm 0.74$ | $4.50 \pm 0.83$ |
| **CAVIA-100** | $1.76 \pm 0.47$ | $1.99 \pm 0.65$ | $4.17 \pm 0.73$ | $4.10 \pm 0.65$ | $4.27 \pm 0.68$ | $4.18 \pm 0.76$ |
| **FLASHCAVIA-100** | $0.002 \pm 0.002$ | $0.005 \pm 0.003$ | $0.005 \pm 0.008$ | $0.014 \pm 0.025$ | $0.003 \pm 0.002$ | $0.008 \pm 0.004$ |
| **NCF** | $0.002 \pm 0.003$ | $0.123 \pm 0.190$ | $0.008 \pm 0.009$ | $0.112 \pm 0.180$ | $0.049 \pm 0.016$ | $0.071 \pm 0.068$ |

Our findings indicate that the extended inner optimization process in FlashCAVIA significantly enhances its performance, unlike the original CAVIA and MAML. However, Fig. 6 suggests that FlashCAVIA may not necessarily benefit from the CSM process, particularly with $d_\xi = 50$. This contrasts the relation NCF has with $d_\xi$. We also observe that NCF exhibits greater overfitting than GBML in **low $N$ - low $|B|$** regimes. These results suggest that for curve-fitting problems permitting large batches of $|B|$, NCF is preferable, while FlashCAVIA should be chosen over other GBML approaches when time and computational resources are abundant.

Given the substantial improvements demonstrated by FlashCAVIA over CAVIA, our subsequent comparisons will focus on the former. Additionally, as the sine regression experiment does not test the model out-of-distribution (OoD), we have designed experiments to explicitly leverage different distributions for training and adaptation.

## 3.2 OPTIMAL CONTROL

While CAVIA and MAML have proven essential for decision-making systems like Meta Reinforcement Learning, NCF's efficacy in this domain remains unexplored. In the scientific machine learning community (Cuomo et al., 2022), optimal control has been investigated as a decision problem, albeit often limited to controlling Neural ODEs to a single target (Chen et al., 2018; Böttcher et al., 2022; Chi, 2024). Neural ODEs offer the implicit benefit of regularizing the control energy (Böttcher & Asikis, 2022). Our experiment in this section aims to evaluate the suitability of NCF and its CSM and $i$CSM mechanisms for optimal control across multiple targets using Neural ODEs.

We seek to leverage the parametrized control signal $\mathbf{u}_\theta$ to drive a 2-dimensional linear ordinary differential equation to a terminal state $\mathbf{x}(T)$. The system is defined as:

$$\begin{cases} \dfrac{d\mathbf{x}^e}{dt}(t) = A\mathbf{x}^e(t) + B\mathbf{u}_\theta(t, \mathbf{x}_0, \xi^e) \\ \mathbf{x}^e(0) = \mathbf{x}_0 \end{cases}, \quad t \in [0, T], \tag{6}$$

where $A = \left(\begin{smallmatrix} 0 & 1 \\ 1 & 0 \end{smallmatrix}\right)$, $B = \left(\begin{smallmatrix} 1 \\ 0 \end{smallmatrix}\right)$, and $\mathbf{x}_0$ is an initial condition sampled from $\mathcal{U}\{-1, 1\}$. The desired target states are denoted $\{\mathbf{x}_*^e\}_{e=1}^M$, each corresponding to one environment or task. We evolve the system and minimize the loss $\ell(\theta, \xi^e, \mathbf{x}_0) = \|\mathbf{x}^e(T) - \mathbf{x}_*^e\|_2^2$.

In all environments, we generate the same $M_{\text{tr}} = 12$ and $M_{\text{tr}} = 1$ initial conditions for meta-training and meta-testing, respectively. Evaluation in both cases is performed on $M_{\text{te}} = 32$ other initial conditions, all stemming from the same underlying distribution. We then proceed to generate $N = 10$ target positions sampled from $\mathcal{U}\{-1, 1\}$ during training, then $N = 16$ targets from $\mathcal{U}\{-2, 2\}$ for meta-testing. Sample initial and target states are reported in Appendix B.4.

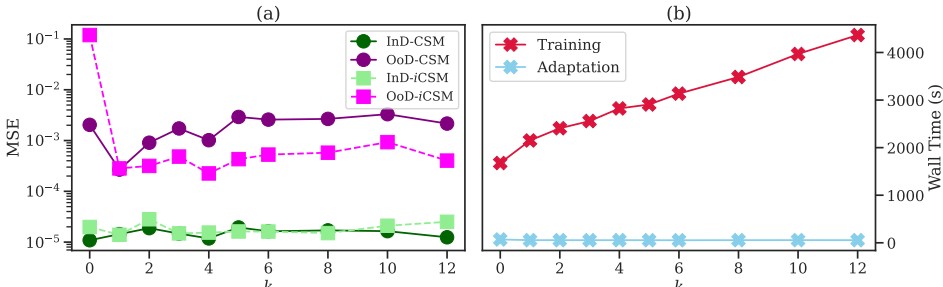

Figure 2: Results on the optimal control experiment with the NCF framework. (a) MSE showing best OoD performance for Taylor order $k = 1$. (b) Training and adaptation times average across CSM and $i$CSM. While training grows linearly up to 4500 seconds (see Section 2), the adaptation time remains constant at roughly 40 seconds (see Table 1), except for $k = 0$.

The results of this experiment, presented in Fig. 2, demonstrate the efficacy of the CSM mechanism across a range of Taylor orders $k$. While maintaining a consistent InD loss, the CSM mechanism shows clear benefits for OoD evaluation, particularly in infinite-dimensional settings. Notably, we observe a linear scaling of wall-clock training time, as predicted by Bettencourt et al. (2019) (see Section 2). The meta-testing time – with CSM deactivated – is near-constant and negligible compared to training time, enabling rapid adaptation as required of any meta-learning method (Hospedales et al., 2021). Additionally, Fig. 2b reveals a slightly higher adaptation time for $k = 0$, suggesting that employing Taylor expansion during training (only to be removed during adaptation) induces regularization of the vector field itself, leading to faster numerical integrations during adaptation.

Although these results appear to unanimously favor $i$CSM based on superior performance and lower parameter count (see Appendix B.4), it is important to note that the lowest OoD performance with CSM is achieved when $k = 1$, before worsening with increased $k$. This observation underscores the importance of aligning model and problem biases; in this case such inductive bias referring to the relation between the control and the target state.

### 3.3 FORCED PENDULUM

The forced pendulum experiment, conducted in a low data-regime similar to (Nzoyem et al., 2024), focuses on learning to reconstruct a dynamical system with varying parameter values. In this case, the vector field to be learned is that of a simple pendulum with a variable forcing functions, where each forcing term represents a distinct environment.

For each environment $e$, we generate multiple trajectories over $t \in [0, 6\pi]$ following the ODE

$$\frac{\mathrm{d}x^e(t)}{\mathrm{d}t} = v^e(t), \quad \frac{\mathrm{d}v^e(t)}{\mathrm{d}t} = -2 \cdot \mu \cdot \omega \cdot v^e(t) - \omega^2 \cdot x^e(t) + F^e(t),$$

where $x^e(t)$ represents position, $v^e(t)$ velocity, $\omega$ the natural frequency, $\mu$ the damping coefficient, and $F^e(t)$ the forcing function. For training environments, we employ 8 oscillating forcing functions with constant or increasing amplitude, while for adaptation, we use 6 functions with faster increasing amplitude. The complete list of forcing terms is provided in Appendix B.5. Support sets comprise 4 initial conditions sampled from $\mathcal{U}\{0, 1\}$ for meta-training and 1 for adaptation, with query sets containing 32 initial conditions. All trajectories are generated using a 4th order Runge-Kutta scheme with $\Delta t = 0.1$. For training NCF, we parametrise our vector field as in Eq. (1), then we implement the CoDA baseline (Kirchmeyer et al., 2022) with exponential scheduled sampling (Bengio et al., 2015) and a context size of $256^4$ (the same value used for CSM within NCF).

---

[4]We note that using a "low-rank" context as suggested in (Kirchmeyer et al., 2022) did not converge. Using $d_\xi = 256$ leads to a massive increase in the total number of parameters, but allows for a more fair comparison.

Our findings, illustrated in Fig. 3, highlight the excellent performance of NCF on this task. In contrast to CoDA, which often becomes trapped in local minima during a significant portion of its training, NCF, whether employing CSM or $i$CSM, consistently improves with increased compute. These differences are further quantified in Table 3, where we observe superior performance with CSM and $i$CSM. Notably, OoD performance is optimized with $i$CSM and $k = 3$, indicating that this infinite-dimensional task benefits from higher-order Taylor approximations of the vector field.

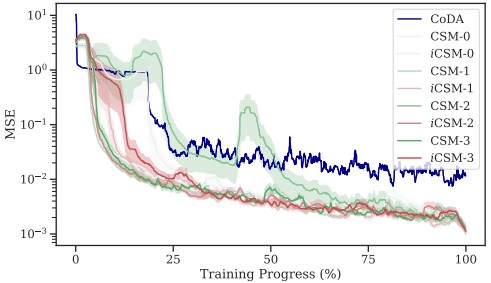

Figure 3: Training losses for the forced pendulum, involving NCF+CSM, NCF+$i$CSM and the baseline CoDA. CSM-$k$ and $i$CSM-$k$ refer to methods with a Taylor order of $k$.

|  | **InD** | **OoD** |
|---|---|---|
| CoDA | $11.2 \pm 4.75$ | $17.2 \pm 6.56$ |
| CSM-0 | $0.79 \pm 0.21$ | $1.75 \pm 0.85$ |
| $i$CSM-0 | $1.20 \pm 0.15$ | $1.86 \pm 0.35$ |
| CSM-1 | $1.08 \pm 0.37$ | $1.66 \pm 0.54$ |
| $i$CSM-1 | $1.42 \pm 0.54$ | $2.09 \pm 0.72$ |
| CSM-2 | $1.18 \pm 0.39$ | $2.01 \pm 1.27$ |
| $i$CSM-2 | $1.39 \pm 0.38$ | $3.48 \pm 3.73$ |
| CSM-3 | $0.98 \pm 0.23$ | $1.91 \pm 0.18$ |
| $i$CSM-3 | $0.96 \pm 0.23$ | $1.48 \pm 0.42$ |

Table 3: MSE results for the forced pendulum. Values are reported in units of $10^{-2}$, with the standard deviation across 3 runs with different seeds. The best results are shaded in grey.

### 3.4 Image Completion

To assess the impact of bi-level optimization schemes like FlashCAVIA and alternating ones like NCF, we focus on the challenging image completion task from (Garnelo et al., 2018; Zintgraf et al., 2019). Using the CelebA32 dataset (Liu et al., 2018), our objective is to learn the mapping $f : [0, 1]^2 \to [0, 1]^3$ from pixel coordinates to RGB values. We treat each image as an environment, meta-training on the CelebA Training split and meta-testing on both its Validation (not reported) and Test splits. The support set for each environment consists of a few $K = 10, 100$ or $1000$ labeled pixels, while the query set comprises all 1024 pixels. For this task, we introduce NCF*, a variation of NCF that eliminates the costly proximal alternating gradient descent regularization mechanism and performs **joint** optimization of both contexts and model weights.

Fig. 4 illustrates that Taylor expansion has a smoothing or underfitting effect, particularly noticeable with FlashCAVIA. Contrasting with additional results in Table 6, we observe that visual quality tends to be negatively correlated with the FlashCAVIA MSE metrics, a sign of overfitting on the few-shot pixels, an effect compounded by the algorithmic adjustments detailed in Appendix A. Additionally, Fig. 4 demonstrates the degradation of results for NCF and NCF* with increasing learning shots, potentially explained by massive overfitting and monotone learning (Bousquet et al., 2022). In their current state, the joint and alternating optimization schemes appear suboptimal for this task, with results falling short of the state-of-the-art (Garnelo et al., 2018). Consequently, further investigations into their generalization capabilities across low and high data regimes are warranted.

## 4 Discussion

### 4.1 Results synthesis

The four experiments conducted in this study collectively establish Contextual Self-Modulation (CSM) (along with its $i$CSM and stochastic variants), as a versatile regularization framework for meta-learning and generalization to unseen environments. This finding aligns with the requirements R1 (task modality), R2 (task dimensionality), and R3 (data regime) outlined in Section 1. When incorporated into FlashCAVIA, CSM exhibits intriguing smoothing properties. We posit that this behaviour may extend to CSM embedded into other contextual meta-learning methodologies, warranting further investigation.

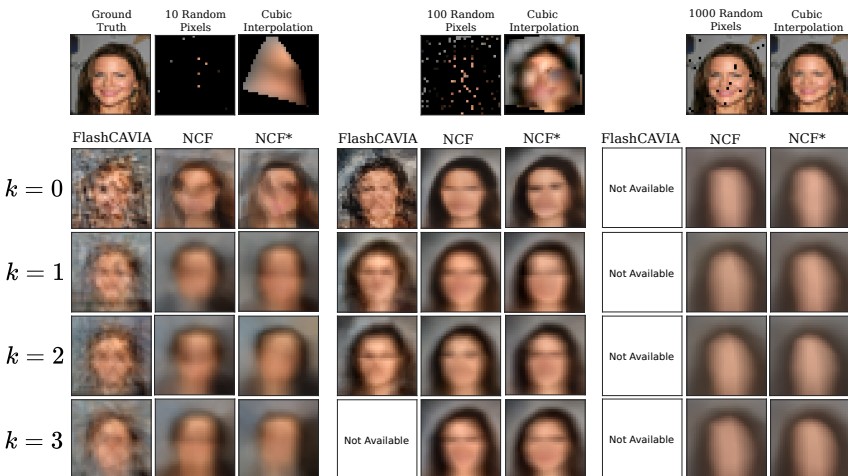

Figure 4: Visual comparison of completed images with varying numbers of random pixels $K$. The first three columns correspond to $K = 10$, the next three to $K = 100$, and the final three to $K = 1000$. The various methods outperform the cubic interpolation baseline up to the highest data regime.

Despite its proximal, stochastic, and CSM regularization mechanisms, our experiments on sine regression and image completion reveal limitations in Neural Context Flows (NCF) for high-data regimes. In these scenarios, NCF and its variants struggle to discriminate between environments amid vast data quantities. With higher-order Taylor expansions, we effectively hit diminishing returns. Conversely, NCF demonstrates unparalleled efficacy in physical systems such as linear optimal control and forced pendulum, presumably due to the inherent regularity these systems exhibit compared to computer vision challenges. This dichotomy suggests that NCF is optimally suited for physical systems with clear inter-task commonalities.

## 4.2 LIMITATIONS

While our work provides extensive experimental results on CSM, several limitations merit acknowledgment • **(i)** First, while substantive discussion of the Neural Process family is provided in Appendix A, our investigation still focuses on Neural Context Flows and Gradient-Based Meta-Learning, and does not encompass *all* contextual meta-learning frameworks • **(ii)** Furthermore, our study did not address classification tasks, which have traditionally served as fertile ground for meta-learning experimentation • **(iii)** Finally, in scenarios precluding the use of forward- or Taylor-mode AD (e.g., Neural ODEs within FlashCAVIA), reliance on reverse-mode AD renders Taylor approximations of order greater than 3 computationally prohibitive. Focusing on FlashCAVIA, we note that further theoretical work is required to elucidate its benefits.

## 4.3 CONCLUSION & FUTURE WORK

This study has substantially advanced the application scope of Contextual Self-Modulation (CSM) beyond Neural Context Flows (NCF), elucidating its efficacy and constraints across a spectrum of tasks and modalities. Our contributions, encompassing the introduction of $i$CSM for infinite-dimensional tasks and StochasticNCF for improved scalability in high-data regimes, offer valuable methodologies for advancing meta-learning in dynamical systems, computer vision challenges, and curve fitting problems. Through extensive empirical evaluation, we have demonstrated $i$CSM's capacity to facilitate OoD generalization and, when integrated with bi-level optimization schemes, to enhance prediction quality. These findings underscore the efficacy of $i$CSM for smooth dynamical systems, where StochasticNCF exhibits superior performance. We also identified several limitations associated with its alternating optimization scheme, notably a low expressiveness in high-data settings. These constraints delineate critical areas for improvement and future research, potentially culminating in a general-purpose framework for meta-learning across a broader range of domains.

ETHICS STATEMENT

The potential adaptability of our meta-learning research to unintended scenarios poses ethical concerns. To mitigate these risks, we commit to rigorous evaluation and validation of our models in diverse, controlled settings prior to their release to the open-source community. This proactive approach aims to ensure responsible development and deployment of our technologies, balancing innovation with ethical considerations.

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

# A   EXTENDED RELATED WORK

This section expands upon Section 1.2 by providing a detailed description of each contextual meta-learning method involved in this work. We elucidate their functionalities, strengths, and weaknesses, as well as their potential to complement and enhance one another.

Table 4: Summary of the major acronyms, definitions, and their original references if traceable, as used throughout this work.

| ACRONYM | DEFINITION | REFERENCE |
|---|---|---|
| InD | In-Domain | |
| OoD | Out-of-Distribution | |
| CSM | Contextual Self-Modulation | (Nzoyem et al., 2024) |
| NCF | Neural Context Flow | (Nzoyem et al., 2024) |
| CNP | Conditional Neural Process | (Garnelo et al., 2018) |
| GBML | Gradient-Based Meta-Learning | |
| MAML | Model-Agnostic Meta-Learning | (Finn et al., 2017) |
| CAVIA | | (Zintgraf et al., 2019) |
| AD | Automatic Differentiation | |

**Neural Process Family.**   Conditional Neural Processes (CNPs) (Garnelo et al., 2018), the progenitors of this family, ingeniously combine the test-time flexibility of Gaussian Processes with the scalability and expressivity of Neural Networks. Utilizing an encoder network $g_\phi$, CNPs construct a permutation-invariant representation $\xi^e$ for each environment in $\mathcal{D}_{\text{tr}}^e$. Both input datapoints $x$ and their corresponding labels $y$ are fed into $g$, adhering to the Deep Sets theory (Zaheer et al., 2017). A decoder $f_\theta$ is then introduced, whose predictions (conditional means and standard deviations) on $(x, \cdot) \in \mathcal{D}_{\text{test}}^e$ are parametrized by $\xi^e$. Trained by minimizing the resulting negative Gaussian probability, CNPs demonstrate significant advantages in function regression, image completion, and dynamics forecasting (Norcliffe et al., 2021). Their particular appeal lies in their suitability for streaming data, given their adaptation complexity scaling as $O(n + m)$, where $n$ and $m$ represent the number of points in $\mathcal{D}_{\text{tr}}^e$ and $\mathcal{D}_{\text{te}}^e$, respectively (cf. Table 1).

While CNPs eliminate the need for gradient updates at test time, they tend to underfit. Extensive research has been conducted to address this weakness, including notable contributions from (Gordon et al., 2019) and (Bruinsma et al., 2023). However, this issue remains a topic of significant interest. Another limitation of NPs is their restrictive assumption of finite-dimensional latent variables, a problem addressed by (Gordon et al., 2019) through the introduction of infinite-dimensional latent variables for translation equivariance, and further explored by (Lee et al., 2023), among others.

**Gradient-Based Meta-Learning.**   Gradient-Based Meta-Learning (GBML) offers an alternative approach to meta-learning through bi-level optimization. These methods optimise shared knowledge $\theta$ in an outer loop while adapting to each task in an inner loop. Model-Agnostic Meta-Learning (MAML) (Finn et al., 2017), for instance, searches for an optimal initialization in the outer loop to facilitate fine-tuning in the inner loop, necessitating the use of second-order derivatives during meta-training. While agnostic to model architecture, this approach scales poorly as models grow larger, a challenge addressed by CAVIA (Zintgraf et al., 2019) whose training algorithm is presented in Algorithm 1[5]. CAVIA leverages external context parameters $\xi := \{\xi^e\}_{e=1}^N$ to modulate the model, with these contexts being the sole parameters adapted during meta-testing. Although more efficient than MAML, CAVIA still requires Hessian information in its bi-level approach to optimising $\{\theta, \xi\}$.

It is important to compare this form of contextual meta-learning with other, more computationally efficient styles, such as joint or alternating optimization. Moreover, GBML methods are susceptible to overfitting, necessitating further investigation into the role of dataset size in the training process. Notably, significant efforts have been made to develop first-order MAML variants, as exemplified

---

[5]Our definition of CAVIA differs from Zintgraf et al. (2019) in that we perform meta-updates on the support sets $\mathcal{D}_{\text{tr}}$, rather than on $\mathcal{D}_{\text{te}}$. This way, CAVIA sees the same labelled datapoints as NCF during its meta-training process (cf. Algorithm 2). This ensures a fair comparison against NCF and CNP.

by iMAML and Reptile (Nichol, 2018), with the latter even outperforming MAML in transductive classification settings. However, the benefits of second-order information remain invaluable for the expressivity of contextual methods like CAVIA, making the improvement of scalability while maintaining accuracy a vital, unsolved challenge. The memory and computational complexity of CAVIA's adaptation rule is presented in Table 1.

**Neural Context Flows.** The original formulation of Neural Context Flow (NCF) (Nzoyem et al., 2024) illustrated in Algorithm 2, optimizes model weights and contexts in an *alternating* manner, diverging from the bi-level optimization approach. This training scheme mirrors the Multi-Task Learning (MTL) joint training paradigm (Wang et al., 2021). To enable effective modulation of weights, NCF introduces the concept of CSM, facilitating seamless information flow across environments. This approach has been successfully tested on several physical systems with limited number of trajectories. NCF offers several advantages, including interpretability and uncertainty quantification, demonstrating state-of-the-art performance against CAVIA (Zintgraf et al., 2019) and CoDA (Kirchmeyer et al., 2022) in few-shot learning of physical systems across dozens of environments. Moreover, for linearly parameterized systems, the underlying physical parameters can be recovered using a simple linear transform (Blanke & Lelarge, 2024). However, the efficacy of CSM in high-data regimes remains largely unexplored.

Comparative studies between GBML and MTL with multi-head structures by Wang et al. (2021) reveal that, in addition to sharing similar optimization formulations in certain settings, the predictions generated by these two methods are comparable, with the gap inversely proportional to neural network depth. Furthermore, their research demonstrates that MTL can achieve similar or even superior results compared to powerful GBML algorithms like MetaOptNet (Lee et al., 2019), while incurring significantly lower computational costs. In our work, we empirically evaluate the performance of these training regimes across various settings.

---

**Algorithm 1** CAVIA Meta-Training

1: **Input:** $\mathcal{D}_{\text{tr}} := \{\mathcal{D}_{\text{tr}}^e\}_{e=1}^N$ defined by $p_{\text{tr}}$
2: $\theta \in \mathbb{R}^{d_\theta}$ randomly initialized
3: $q_{\max}, |B|, H \in \mathbb{N}^*; \eta_\theta, \eta_\xi > 0$
4: **for** $q \leftarrow 1, q_{\max}$ **do**
5:     Sample batch of $|B|$ tasks from $p_{\text{tr}}(\mathcal{E})$
6:     **for** $e \leftarrow 1, |B|$ **do**
7:         $\xi^e = \mathbf{0}$
8:         **for** $h \leftarrow 1, H$ **do**
9:             $\xi^e = \xi^e - \eta_\xi \nabla_\xi \mathcal{L}^e(\theta, \xi^e, \mathcal{D}_{\text{tr}}^e)$
10:         **end for**
11:     **end for**
12:     $\xi := \{\xi^e\}_{e=1}^{|B|}$
13:     $\theta = \theta - \eta_\theta \nabla_\theta \mathcal{L}(\theta, \xi, \mathcal{D}_{\text{tr}})$    ▷ Eq. (4)
14: **end for**

**Algorithm 2** NCF Meta-Training

1: **Input:** $\mathcal{D}_{\text{tr}} := \{\mathcal{D}_{\text{tr}}^e\}_{e=1}^N$
2: $\theta_0 \in \mathbb{R}^{d_\theta}$ randomly initialized
3: $\xi_0 := \{\xi^e\}_{e=1}^N$, where $\xi^e = \mathbf{0} \in \Xi$
4: $q_{\max} \in \mathbb{N}^*; \beta \in \mathbb{R}^+; \eta_\theta, \eta_\xi > 0$
5: **for** $q \leftarrow 1, q_{\max}$ **do**
6:     $\mathcal{G}(\theta) := \mathcal{L}(\theta, \xi_{q-1}, \mathcal{D}_{\text{tr}}) + \frac{\beta}{2}\|\theta - \theta_{q-1}\|_2^2$
7:     $\theta_q = \theta_{q-1}$
8:     **repeat**
9:         $\theta_q \leftarrow \theta_q - \eta_\theta \nabla \mathcal{G}(\theta_q)$
10:     **until** $\theta_q$ converges
11:     $\mathcal{H}(\xi) := \mathcal{L}(\theta_q, \xi, \mathcal{D}_{\text{tr}}) + \frac{\beta}{2}\|\xi - \xi_{q-1}\|_2^2$
12:     $\xi_q = \xi_{q-1}$
13:     **repeat**
14:         $\xi_q \leftarrow \xi_q - \eta_\xi \nabla \mathcal{H}(\xi_q)$
15:     **until** $\xi_q$ converges
16: **end for**

---

In conclusion, modulating neural network behaviour with contextual information presents a non-trivial challenge. Beyond CNPs, CAVIA, and NCF, numerous approaches have attempted this task using global context vectors (Norcliffe et al., 2021; Massaroli et al., 2020) and other methodologies. The current landscape of contextual meta-learning is fragmented, with some methods tailored for specific regression tasks, others for physical systems (Day et al., 2021; Nzoyem et al., 2024) or vision challenges (Zintgraf et al., 2019; Garnelo et al., 2018), and many barely accommodating classification tasks (Norcliffe et al., 2021). This fragmentation underscores the necessity for our work, with the conclusions drawn herein aiming to elucidate and advance this rapidly evolving field.

# B ADDITIONAL DETAILS & RESULTS

This section provides supplementary information on model architectures and training configurations used in our experiments. We also present additional results that enhance the interpretations and conclusions reached in the main section. Throughout our experiments, we employ multilayer perceptrons

(MLPs) with Swish activation functions unless otherwise specified. The Adam optimizer (Diederik, 2014) with a constant learning rate is used as the default for model weight optimization. For the (infinite-dimensional) context vectors, the choice of optimizer depends on the method: Adam with a learning rate of $10^{-3}$ for all Neural Context Functions (NCF) experiments, and stochastic gradient descent (SGD) with a learning rate of $10^{-3}$ for all FlashCAVIA experiments (unless otherwise specified). The hyperparameters for the original CAVIA and MAML follow the conventions established in (Zintgraf et al., 2019). When dealing with dynamical systems, we utilize the Dopri5 integrator (Kidger, 2022) by default, with datasets following the interface described in the Gen-Dynamics(**?**) initiative (Nzoyem, 2024). The CoDA implementation for dynamical systems relies on the library by Chen (2018). For methods in this work, we used a RTX 4080 GPU to accelerate the training.

## B.1 LOTKA-VOLTERRA

We conduct an experiment with FlashCAVIA using the $i$CSM mechanism on the Lotka-Volterra problem, as described in (Kirchmeyer et al., 2022). This problem is chosen for its well-understood linearity and its relevance to the NCF task. The Lotka-Volterra dynamics describe the evolution of prey ($x$) and predator ($y$) populations over time ($t$):

$$\frac{\mathrm{d}x}{\mathrm{d}t} = \alpha x - \beta xy,$$

$$\frac{\mathrm{d}y}{\mathrm{d}t} = \delta xy - \gamma y,$$

where $\alpha$ represents the prey population growth rate, $\beta$ the predation rate, $\delta$ the rate at which predators increase by consuming prey, and $\gamma$ the natural death rate of predators. The time-invariant parameters $\beta$ and $\delta$ define our 9 meta-training and 4 meta-testing environments, as described in (Kirchmeyer et al., 2022).

We set our model hyperparameters identical to those in (Nzoyem et al., 2024), with the context function implemented as a 3-layer MLP with 32 hidden units and 128 output units. To evolve our dynamics through time in a differentiable manner, we leverage a custom RK4 integrator based on JAX's `scan` primitive (Bradbury et al., 2018).

In this work, we employ $i$CSM and observe that the predictions of the context functions are themselves invariant with time, mirroring the time-invariant nature of the two parameters $\beta$ and $\delta$ they are meant to encode (see Fig. 5). This observation, along with results in Section 3.2, suggests that the $i$CSM protocol generalizes the CSM and should be prioritized for maximum flexibility and performance.

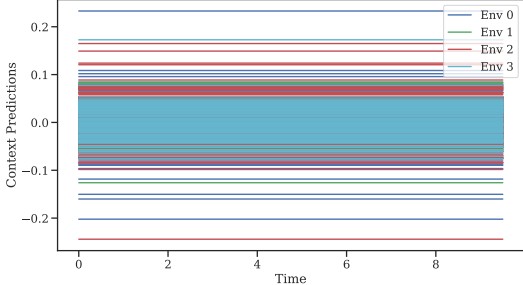

Figure 5: Predictions from the infinite-dimensional context functions on the Lotka-Voltera dynamical problem after adaptation to 4 environments. This case uses the FlashCAVIA method. Since this is a finite-dimensional problem, the model learns to ignore the input $t$ entirely and predict constant values (albeit different for different environments).

## B.2 DIVERGENT SERIES

This experiment is designed to investigate the effect of diverging power series used in the data generating process on the performance of CSM. We generate trajectories with two carefully crafted

vector fields $f$, where the varying parameters $c$ are spaced farther than $f$'s convergence radius $R$ (with respect to the parameter $c$). The first experiment sets $f : x \mapsto \frac{x}{1-c}$ with $R = 1$, while the second experiment sets $f : x \mapsto \frac{x}{1+(cx)^2}$ with $R = \frac{1}{|x|}$, meaning the suitability of a context $\xi^e$ for Taylor approximation near $\xi^j$ depends on the system's state.

For meta-training in both experiments, we select 7 values for $c \in [1.25, 10)$ regularly spaced by 1.25. For meta-testing, we similarly choose 15 regularly spaced values in $[1.25, 20)$. The model is trained for 100 outer steps with 10 inner steps in the NCF proximal alternating minimization. We place 4 contexts in the context pool, each with finite dimensionality $d_\xi = 128$. Adaptation is performed for 7500 steps, still resulting in a fraction of the total training time. The initial learning rate for the Adam optimizer for both weights and contexts is set to $5e - 4$, with a scheduling factor of 0.5 for scaling at one and two-thirds of the total 2000 training steps.

We use the same model architecture as the previous experiment for both data and main networks. The context network (embedded in the vector field) is a 1-layer MLP with 128 input units, 32 hidden units, and 128 output units. We employ 10 inner steps and 100 outer steps in the NCF proximal algorithm. Our results, presented in Table 5, demonstrate that the CSM mechanism can successfully reconstruct the trajectories despite the use of extremely small neural networks. This finding underscores the need for a clearer notion of task-relatedness in dynamical systems based on power series and radii of convergence, which would complement efforts in the meta-learning community to define such notions of relatedness (Khodak et al., 2019).

Table 5: In-Domain (InD) and adaptation (OOD) test MSEs ($\downarrow$) for the divergent series problems. The first ODE, termed ODE-1, is $x \mapsto \frac{x}{1-c}$, and the second, termed ODE-2, is $x \mapsto \frac{x}{1+(cx)^2}$. We additionnaly indicate the small size of the vector field used to learn these prametric mappings.

| | ODE-1 ($\times 10^{-4}$) | | | ODE-2 ($\times 10^{-4}$) | | |
|---|---|---|---|---|---|---|
| | #PARAMS | IND | OOD | #PARAMS | IND | OOD |
| NCF-$t_0$ | 2099 | $0.47 \pm 0.01$ | $1.16 \pm 0.2$ | 2099 | $5.28 \pm 0.12$ | $6.71 \pm 0.41$ |
| NCF-$t_2$ | 2099 | $7.4 \pm 0.25$ | $11.0 \pm 0.67$ | 2099 | $5.34 \pm 0.23$ | $6.84 \pm 0.29$ |

### B.3 SINE REGRESSION

For the sine regression task, we closely followed the directions outlined in (Zintgraf et al., 2019), including the model architecture. However, we made one notable exception: the activation function was switched from ReLU to Softplus to encourage smoothness in the approximation. This modification allows for a more nuanced comparison of the different approaches.

Our training configuration employed $|B|$ tasks per meta-update, which also determines the number of environments contributing to the NCF loss function. To optimize computational efficiency, we processed all datapoints within each environment simultaneously. The number of outer steps or epochs varied by GBML approach, such that the training time was constrained to 6, 10, or 60 minutes for $K = 1, 5$, and 100 inner gradient updates, respectively. NCF, being independent of $K$, was limited to 6 minutes, at which point its loss curve had stabilized. While these time constraints may not yield minimum test error in all cases, they enable fair comparison, particularly as many strategies had reached peak performance within these timeframes.

Controlling the total number of environments used in the training process is straightforward using the `gen-dynamics` interface (Nzoyem, 2024). For the original CAVIA and MAML, we control $N$ by setting two values in the dataloader class that reset each time we reach the threshold of $N$ environments. While the phase and amplitude seed is reset, the input-generating seed is allowed to change to generate diverse datapoints as the training evolves, ensuring a rich and varied dataset.

In our FlashCAVIA implementation, we initially set the inner learning rate to $10^{-3}$. However, this occasionally led to divergences. In such cases, we adjusted the rate to $10^{-4}$, which resolved the issue while maintaining fair comparison. This adjustment is justified as the original CAVIA is documented to scale its gradients with its inner learning rate (Zintgraf et al., 2019).

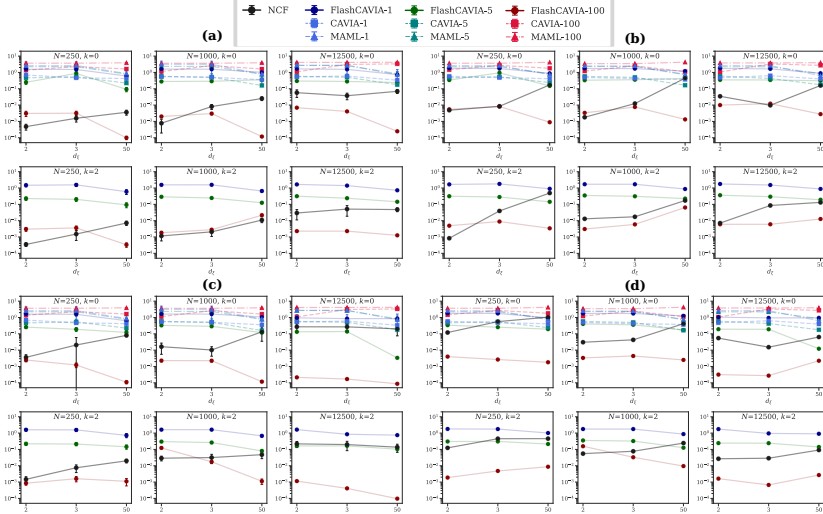

Figure 6: Visualization of test MSEs for MAML, CAVIA, FlashCAVIA, and NCF on the sine regression problem with varying numbers of environments $|B|$ per meta-update: $|B| = 250$ (Large), and $|B| = 25$ (Small). Complementing Table 2, this experiment presents 24 comparisons in groups of 6. (a) Meta-Train Large, (b) Meta-Test Large, (c) Meta-Train Small, (d) Meta-Test Small. The horizontal bars plot the standard deviation across many evaluation environments.

## B.4 OPTIMAL CONTROL

The CSM mechanism was applied to the vector field as prescribed in (Nzoyem et al., 2024). We parametrized **u** as a set of MLPs with a 3-network architecture, utilizing either a latent vector of size $d_\xi = 2$ as context or a 2-network architecture (without the context network, the two others identical) with bespoke small MLPs as latent contexts. During meta-training, the total number of learning parameters ($|\theta| + |\Xi|$) with CSM was 6581, while with $i$CSM it was only 5749. For CSM, we found that small context sizes consistently provided better results for this task. For optimization, we employ differentiable programming, which, despite the extensive literature on adjoint methods for controlling physical systems, has demonstrated significant results for optimal control tasks (Kidger, 2022; Nzoyem et al., 2023).

On this task, we trained the NCF framework for 5000 outer steps with 10 inner steps. Adaptation was also performed for 5000 steps. We employed the same optimizers as in the previous experiment, but with an initial learning rate of $10^{-3}$ and a scaling factor of 0.25. The context pool size was set to 2, matching the context size.

For the network architecture, we designed the data network with an input and an output layer, each containing 32 hidden units and outputs. Before feeding into the data network, we concatenated the input $\mathbf{x}_0$ with $t$. The main network consisted of a 3-layer MLP with 32 hidden units and 1 output unit. When implementing $i$CSM, we maintained the data and main networks while removing the context network. The infinite-dimensional context function was parametrized as a 2-layer MLP with 32 hidden units and 1 output unit. This careful design of the network architecture and training process allowed us to effectively capture the dynamics of the optimal control problem, as evidenced by the results presented in Figures 7 and 8.

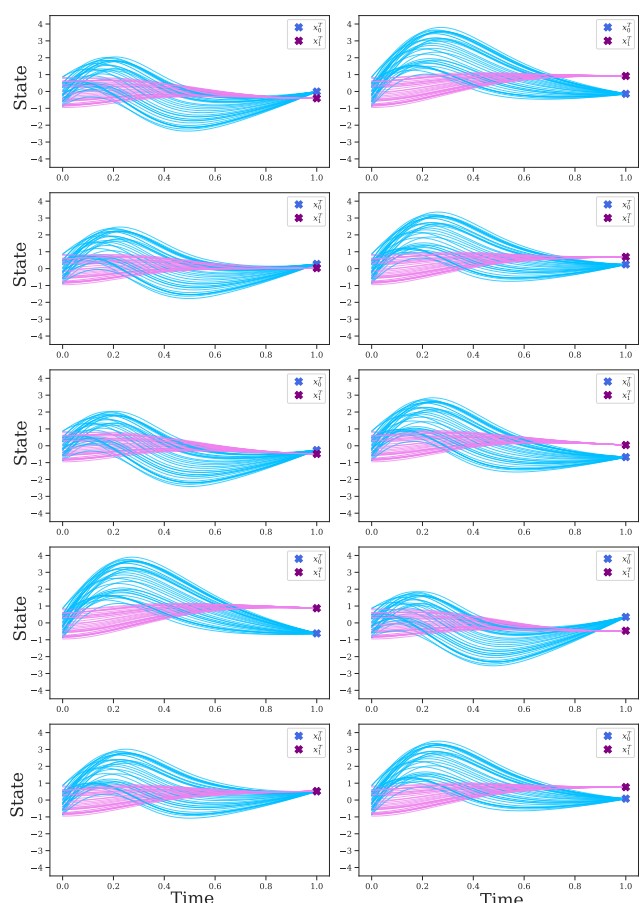

Figure 7: Trajectories during evaluation on the optimal control problem's meta-training query sets. This result is for Taylor order $k = 0$ using NCF with CSM.

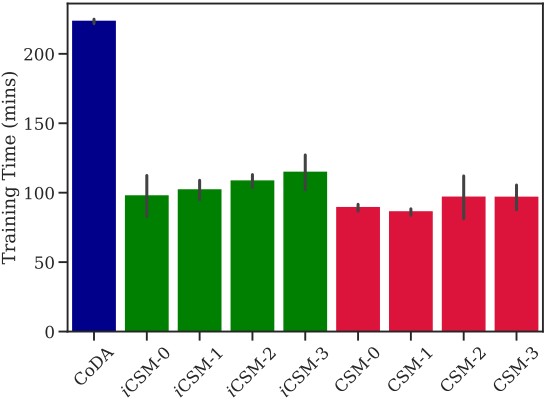

Figure 9: Training times on the forced pendulum problem. We observe a marginal increase in training times with the Taylor order. The vertical bars indicate the standard deviation across 3 runs.

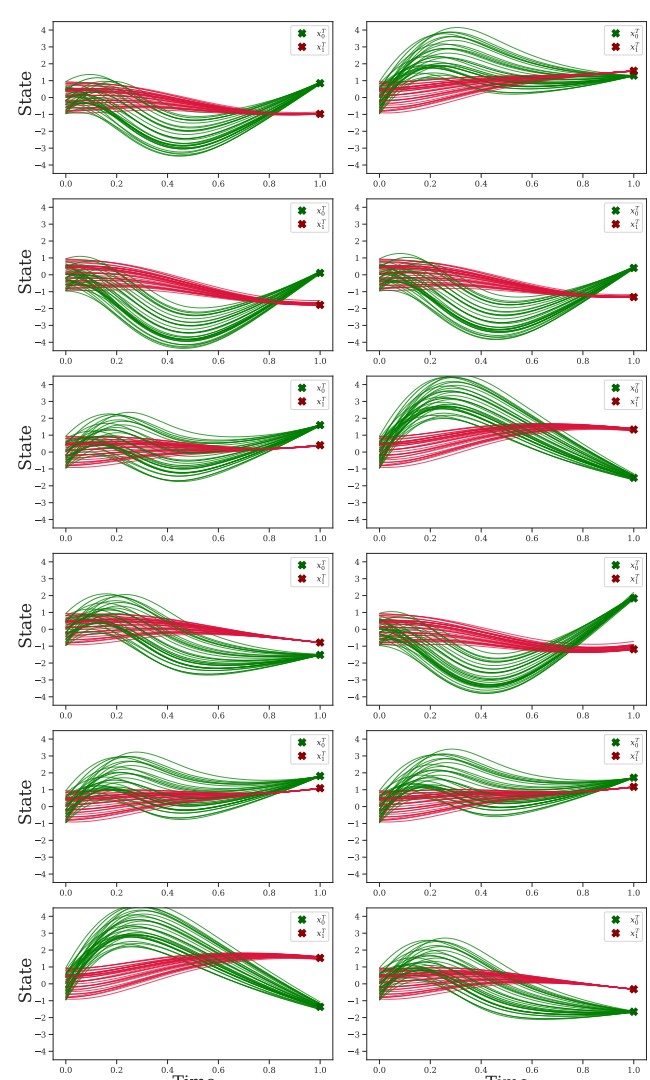

Figure 8: Trajectories during evaluation on the optimal control problem's meta-testing query sets. This result is for Taylor order $k = 0$ using NCF with CSM.

## B.5 FORCED PENDULUM

We parametrized our vector field as in Eq. (1), utilizing either a 3-network or 2-network MLP architecture depending on the use of NCF with CSM or $i$CSM, all making use of Swish activations (Ramachandran et al., 2017). Backpropagation of gradients is performed through the internals of the numerical integrator to minimize the average MSE loss across all trajectory steps. We vary the number of Taylor orders from $k = 0$ to $k = 3$, with a fixed context pool size of $p = 2$. For CSM, we set the context size to $d_\xi = 256$, while for $i$CSM, we use a 2-layer MLP with 32 hidden units and 256 output units. For this task, we solely considered the CoDA baseline (Kirchmeyer et al., 2022). We were unable to run FlashCAVIA due to the impracticality of forward-mode Taylor expansion within its bi-level optimization framework, which includes an integration scheme with custom differentiation rules.

We add that to allow a fair comparison, we ensured that the number of parameters in CoDA root's network was near-identical to that NCF's main network plus state networks (see also (Park et al., 2023) for a similar comparison strategy). It is important to note that only their main/root network's

Table 6: Results for image completion, reported with standard deviation across many Training and Testing evaluation environments. The cell with the lowest MSE in each column is shaded in grey. Taylor order hyperparamter $k$ is reported following the method's name. The cross $\times$ points out experiments that couldn't be run due to memory limitations.

| | $K = 10$ | | $K = 100$ | | $K = 1000$ | |
|---|---|---|---|---|---|---|
| | TRAIN | TEST | TRAIN | TEST | TRAIN | TEST |
| **FLASHCAVIA-0** | $0.0008 \pm 0.0002$ | $0.0987 \pm 0.0378$ | $0.0051 \pm 0.0013$ | $0.1350 \pm 0.0672$ | $\times$ | $\times$ |
| **NCF-0** | $0.0238 \pm 0.0060$ | $0.0493 \pm 0.0205$ | $0.0114 \pm 0.0029$ | $0.0214 \pm 0.0134$ | $0.0378 \pm 0.0095$ | $0.0342 \pm 0.0173$ |
| **NCF\*-0** | $0.0043 \pm 0.0011$ | $0.0466 \pm 0.0186$ | $0.0067 \pm 0.0017$ | $0.0188 \pm 0.0116$ | $0.0387 \pm 0.0097$ | $0.0321 \pm 0.0168$ |
| **FLASHCAVIA-1** | $0.0021 \pm 0.0005$ | $0.1010 \pm 0.0456$ | $0.0118 \pm 0.0030$ | $0.1370 \pm 0.0737$ | $\times$ | $\times$ |
| **NCF-1** | $0.0297 \pm 0.0074$ | $0.0499 \pm 0.0200$ | $0.0169 \pm 0.0042$ | $0.0242 \pm 0.0145$ | $0.0392 \pm 0.0098$ | $0.0353 \pm 0.0164$ |
| **NCF\*-1** | $0.0086 \pm 0.0022$ | $0.0457 \pm 0.0174$ | $0.0127 \pm 0.0032$ | $0.0207 \pm 0.0113$ | $0.0375 \pm 0.0094$ | $0.0322 \pm 0.0164$ |
| **FLASHCAVIA-2** | $0.0019 \pm 0.0005$ | $0.0998 \pm 0.0472$ | $0.0119 \pm 0.0030$ | $0.1360 \pm 0.0740$ | $\times$ | $\times$ |
| **NCF-2** | $0.0259 \pm 0.0065$ | $0.0489 \pm 0.0202$ | $0.0166 \pm 0.0042$ | $0.0240 \pm 0.0144$ | $0.0428 \pm 0.0107$ | $0.0350 \pm 0.0164$ |
| **NCF\*-2** | $0.0095 \pm 0.0024$ | $0.0453 \pm 0.0190$ | $0.0129 \pm 0.0032$ | $0.0212 \pm 0.0118$ | $0.0406 \pm 0.0102$ | $0.0328 \pm 0.0171$ |
| **FLASHCAVIA-3** | $0.0034 \pm 0.0009$ | $0.1030 \pm 0.0486$ | $\times$ | $\times$ | $\times$ | $\times$ |
| **NCF-3** | $0.0294 \pm 0.0074$ | $0.0510 \pm 0.0209$ | $0.0176 \pm 0.0044$ | $0.0236 \pm 0.0145$ | $0.0380 \pm 0.0095$ | $0.0360 \pm 0.0165$ |
| **NCF\*-3** | $0.0086 \pm 0.0022$ | $0.0448 \pm 0.0188$ | $0.0129 \pm 0.0032$ | $0.0211 \pm 0.0122$ | $0.0373 \pm 0.0093$ | $0.0328 \pm 0.0169$ |

parameter counts are intended to match. For NCF, we employ the Adam optimizer (Diederik, 2014) with 2000 outer steps and 10 inner steps. Adaptation is performed for 1500 steps.

Interestingly, we found that NCF can achieve better performance than those presented in Table 3 by using a different set of hyperparameters similar to those in Appendix B.1, but with the Dopri5 scheme. This modification results in an MSE reduction of about one order of magnitude. However, these considerations do not alter our conclusions in any significant way.

The complete list of forcing terms used to generate trajectories in our training environments includes: $\sin(t), \cos(t), \sin(t)+\cos(t), e^{\cos(t)}, \sin(\cos(t)), e^{\cos(t)}, \sin(\sin(t)+\cos(t)), \sinh(\sin(t)+\cos(t)),$ $\sinh(\sin(t)), \sinh(\cos(t)), \tanh(\cos(t))$. For the adaptation environments, we used: $\sinh(\cos(t)),$ $\tanh(\cos(t)), \sin(t) \cdot e^{0.01t}, \cos(t) \cdot \log\left(\frac{t}{10} + 1\right), \sin(t) + \frac{t}{10}, \sin(t) \cdot (1 + 0.02t)$. Despite their clear discrepancies, candidate trajectories from all these environments were processed in a full-batch, and the mean MSE was minimized to expedite training.

## B.6 IMAGE COMPLETION

For the image completion task, we adopted the model hyperparameters from (Zintgraf et al., 2019), implementing a 5-layer MLP with 128 hidden units each and a context size of 128. However, we made one significant modification: the activation function was changed to softplus to promote smoothness in the model's behavior. This alteration allows for potentially more nuanced image completions.

Following Zintgraf et al. (2019), we use an MLP with 5 hidden layers of 128 nodes each, followed by ReLU activations. The context vector is directly concatenated to the 2 inputs before being fed into the first layer of the MLP. We test various orders of Taylor expansion, from $k = 0$ to $k = 3$, for all methods. To ensure fair comparison, we adjust the number of epochs/steps to maintain a consistent training duration of 1.5 hours across all methods.

In our FlashCAVIA implementation, we use 4 inner gradient updates, while for NCF, we employ 20 in the inner approximation of its proximal operators and a fixed 500 adaptation steps. To expedite FlashCAVIA training, we meta-train on $|B| = 512$ environments simultaneously, which unfortunately precluded running FlashCAVIA with high $k$ and/or $K$ values. We set the outer learning rate to $10^{-4}$ and the inner learning rate to $10^{-1}$. For NCF, we used a constant learning rate of $10^{-3}$ for both model weights and contexts. To ensure fair comparison, we kept the number of training steps under control, with no run lasting longer than 1.5 hours.

Our main results, presented in Table 6, demonstrate the performance of these methods on the CelebA Training and Testing splits. FlashCAVIA's performance gradually improves as $K$ increases, aligning with (Zintgraf et al., 2019). For NCF and NCF\*, optimal MSEs are obtained at $K = 100$, with a slight advantage for the less-regularized NCF\*. Interestingly, performance degrades for $K = 1000$ compared to $K = 100$, an observation we found consistent across various batch sizes, model architectures, and other hyperparameters during training.

It is worth noting that both NCF and FlashCAVIA can achieve more accurate results when allowed to train for longer periods. The various plots in this section showcases these improved results, suggesting the potential for even higher quality image completions with extended training time.

TRAIN                              TEST

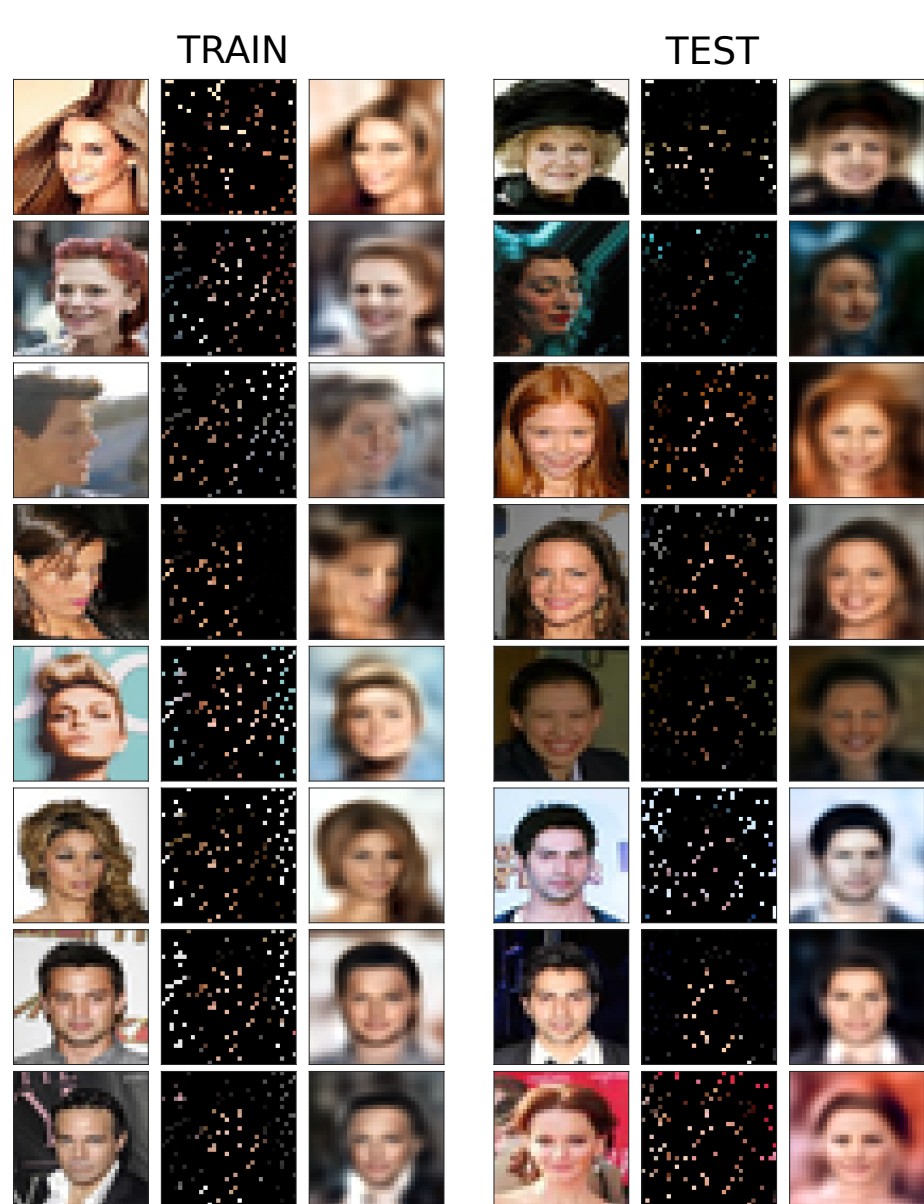

Figure 10: Sample train and test image completions using FlashCAVIA after approximately 60 hours of training. This visualization used 100 random pixels.

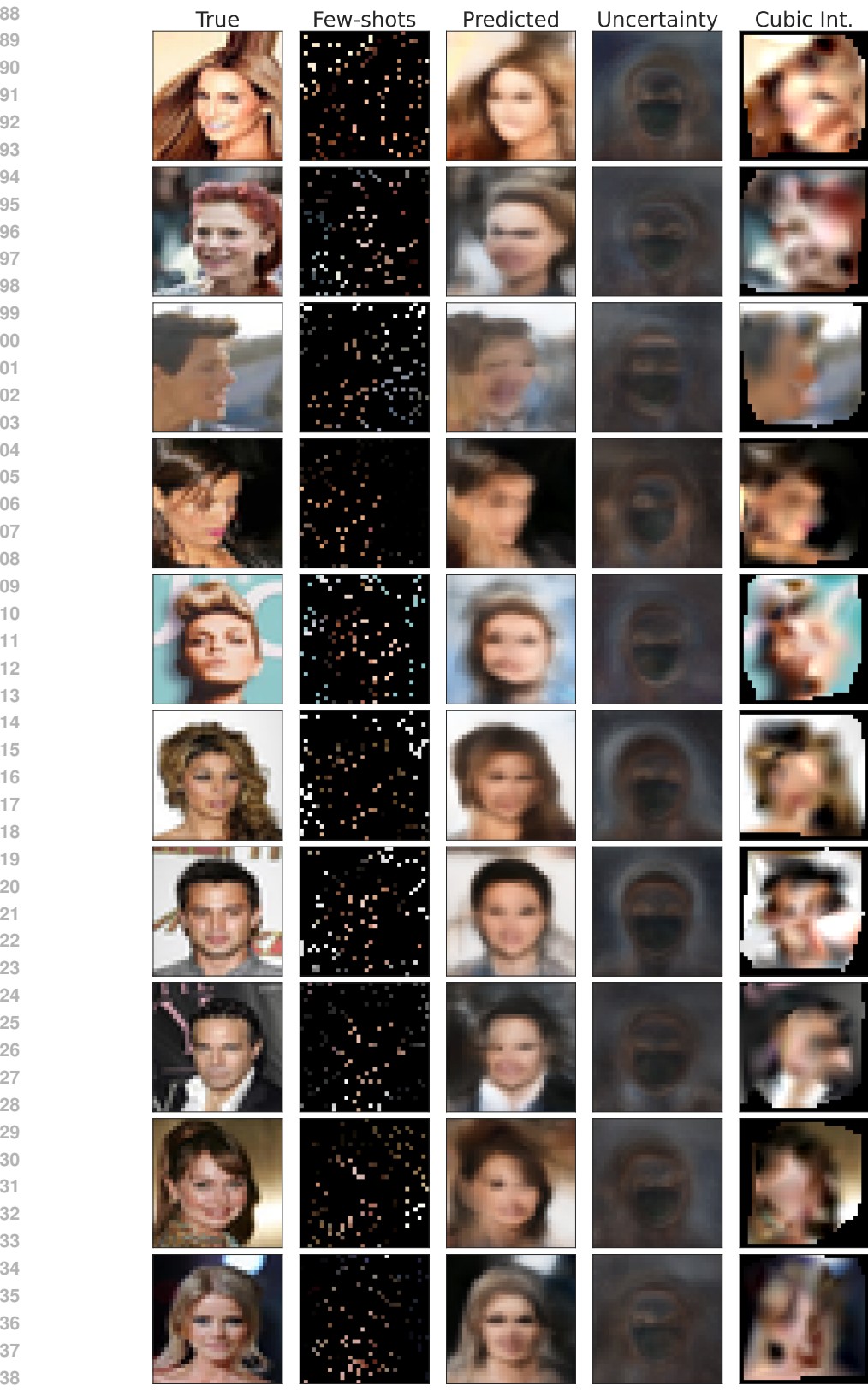

Figure 11: Train-time face completions with NCF* using Taylor order $k = 2$, the same order with which uncertainties (variance across 466 candidate predictions) are calculated. This visualization used 100 random pixels.

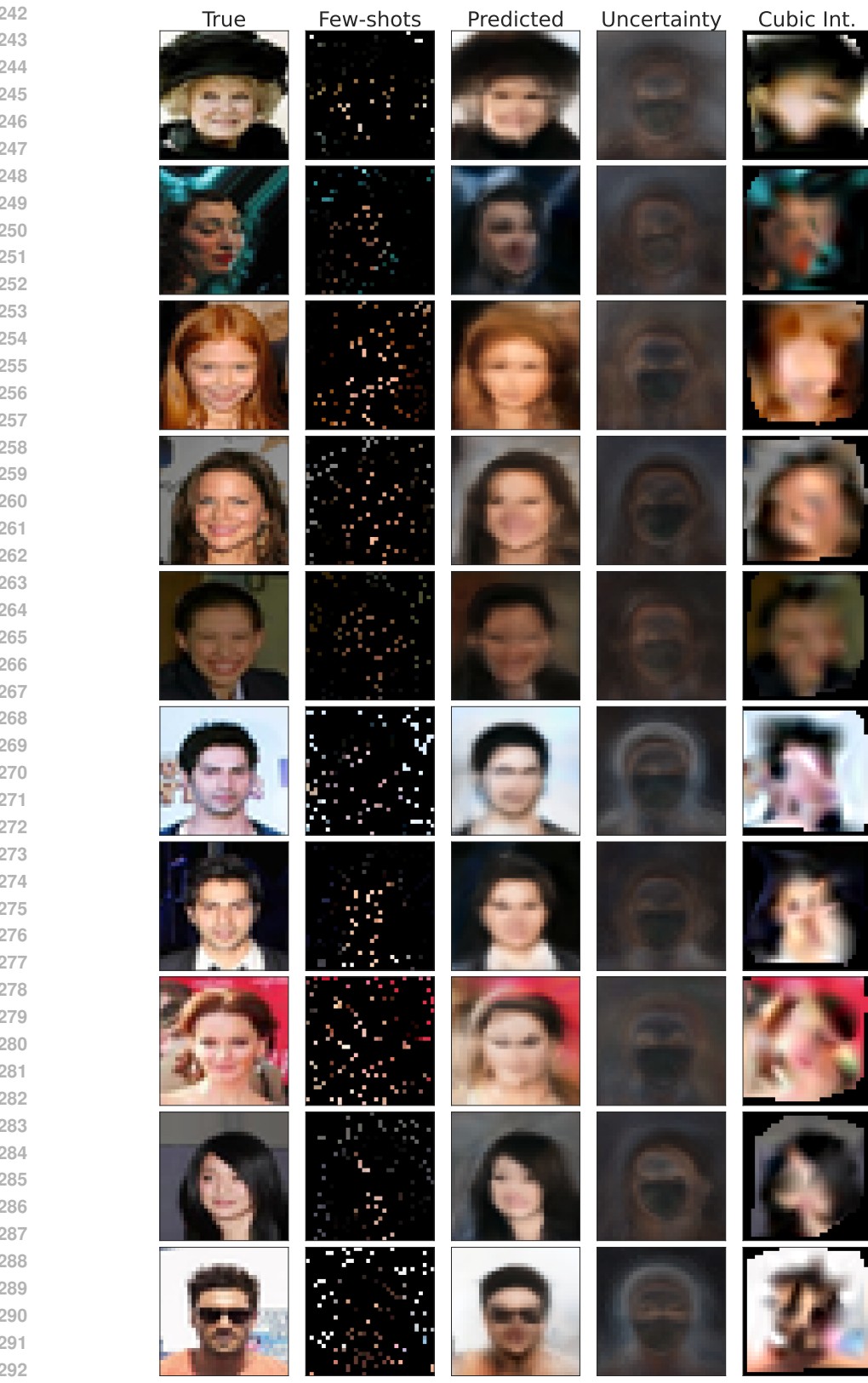

Figure 12: Test-time face completions with NCF* using Taylor order $k = 2$, the same order with which uncertainties (variance across 466 candidate predictions) are calculated. This visualization used 100 random pixels.

