# OpenReview forum: "Extending Contextual Self-Modulation: Meta-Learning Across Modalities, Task Dimensionalities, and Data Regimes"
_ICLR.cc/2025/Conference — Submitted to ICLR 2025_

### Official Review · Reviewer_rKCA · 2024-10-24

**Soundness:** 3
**Presentation:** 1
**Contribution:** 2
**Rating:** 3
**Confidence:** 2

**Summary:**

This paper proposes to extend Contextual Self-Modulation (CSM), which is an uncertainty-handling mechanism of Neural Context Flow (NCF). NCF is a robust meta-learning framework of neural ODE that includes self-modulation with high-order Taylor expansion around contexts. The authors focus on generalizing CSM into high data regimes and designed StochasticNCF and FlashCAVIA, which take CSM in various directions. From comparative experiments, the author verified that the NCF can be successfully extended to various optimization and meta-learning techniques.

**Strengths:**

1. This paper proposed various possibilities of neural context flow that can contribute to studies of meta-learning in dynamical systems.
2. The authors have proposed various efficient and scalable extensions to CSM, which can be comprehended at a high level.
3. The study advanced the applicability of CSM to a wide spectrum of tasks and modalities.

**Weaknesses:**

1. I believe this paper is not clearly written.
    * iCSM: It is briefly mentioned that the space of iCSM is "an infinite-dimensional variation" that utilizes "a space of multi-layer perceptrons, whose weights are flattened into a 1-dimensional tensor." I do not find a good enough explanation, mathematical notation, or relevant resources to understand this concept fully.
    * StochasticNCF: this translates high-order Taylor expansion to standard SGD estimation. Then, Isn't it just another formulation of the first-order Taylor expansion? If it is just a specific case of applying the Taylor expansion, it might not be considered a pure contribution of this work. Moreover, the author should have provided meaningful theoretical or experimental results of this design.
    * FlashCAVIA: FlashCAVIA is claimed as a powerful upgrade of CAVIA, in terms of efficiency and custom optimizer. First, the presentation of CAVIA in the manuscript is not appropriate; for example, I do not understand why specifically CAVIA is chosen among various meta-learning approaches. Also, not much theoretical advances of FlashCAVIA from CAVIA (Algorithm 1) can be found to be an "upgrade." If the "custom optimizer" is one of the key improvements, its benefit must theoretically be validated within the same number of gradient updates.
2. Building on the previous point, the contributions of this paper primarily lie in the implementation of dynamical meta-learning frameworks. While I do believe these technical contributions are important, certain claims on efficiency and scalability require additional experiments for full validation.
3. It is crucial to demonstrate the performance of FlashCAVIA-100, MAML-100, and CAVIA-100, along with the total elapsed time measurements for training.
4. As stated in Section 4.2, some essential benchmarks for meta-learning need to be included.

**Questions:**

Please see the Weaknesses section.

---

> ### Author Response · Authors · 2024-11-25
>
> Thank you for reading our paper and for providing invaluable feedback. We are happy you agree with the wide possibilities of Neural context Flows in meta-learning dynamical systems and several other tasks and modalities. We strived to make our additive improvements understandable on an intuitive level, and we are happy you found the same. Please find below our answers to your concerns.
>
> ---
>
> ### W1. Improvement on the writing
> 1. __Clearer exposition of iCSM__. We thank you for pointing this out. We have updated our description of iCSM, which now includes mathematical notations (__line 210__). We believe this will make the understanding of iCSM and how different it is from CSM clearer.
> 2. __StochasticNCF__. We have equally modified __line 152__ to clearly indicate the motivation for StocashticNCF, which is not theoretical, but a practical one -- we couldn't run any of the high-data regimes experiments unless we implemented StochsticNCF. We have also modified __Section 2.2__ to make this clear.
> 3. __Why CAVIA ?__ We chose CAVIA [1] because it is a __contextual__ meta-learning approach and powerful representative of the gradient-based meta learning family that uses bi-level optimisation. Due to space constraints, we couldn't stress this in the main text, and left it for the __Appendix A__. Our use of the term "upgrade" was based on the visually better performance it offered compared to CAVIA across our experiments. We've now rephrased it to "update", and we modified the corresponding __Section 2.3__ to indicate that the main contribution of FlashCAVIA is that it incorporates CSM, unlike the original CAVIA. Ultimately, FlashCAVIA remains a practical update, but we've added in our intended __Future Work__ that theoretical work is needed to elucidate some of its benefits.
>
> [1] Zintgraf et al. Fast Context Adaptation via Meta-Learning, ICML 2019
> ### W2. Claims on efficiency
> As we mentioned in the previous section, we've clarified our contributions. We've also modified our claims in the paper such that our conclusions are conservative and only limited to the observed variety of experiments we have (See __lines 30-31, 509-510__, etc.)
>
> ---
>
> ### W3. Total elapsed time for training
>
> Our study is focused on the meta-learning accuracy of each framework, and we mostly disregarded training time during. We focused on making sure each method was trained long enough that its loss was optimal. As our since regression experiment can attest, FlashCAVIA-100 is indeed performant, more so than CAVIA-100 and MAML-100.
>
> Thank you for requesting this clarification. Below, we provide the average times we observed for the methods with 100 gradient steps. They are averaged across context sizes $d_{\xi}$ and Taylor orders $k$ as we note in the Sine Experiment's Table 2. We note that both MAML [1] and CAVIA [2] use the reference implementation from [2], compared to our custom implementation for FlashCAVIA. The training times, averaged over Taylor orders and context sizes, are presented below in seconds. FlashCAVIA took roughly 5 mins to train whereas CAVIA took about 23 mins, and finally MAML 28 mins. This increase in performance is due to the __parallelisation__ we mention in our Section 2.3.
>
> | Method         | Time   |
> | -------------- | ------ |
> | FlashCAVIA-100 | 350 s  |
> | CAVIA-100      | 1407 s |
> | MAML-100       | 1701 s |
>
> [1] Finn et al. Model-Agnostic Meta-Learning for Fast Adaptation of Deep Networks, ICML 2017
>
> [2] Zintgraf et al. Fast Context Adaptation via Meta-Learning, ICML 2019
>
> ---
>
> ### W4. Additional benchmarks
> __Section 4.2__ is used to recognize the limitations of our work, which we believe are big enough in their own right to merit a full paper. We thank you and we take this interest for our current limitations as a sign that our work is valuable and should be continued.
>
> ---
>
> Once again, we wish to express all our thanks for your feedback. The modifications we made have clearly improved the clarity, particularly regarding our claims.

---

> > ### Author Response · Authors · 2024-11-30
> >
> > Dear Reviewer,
> >
> > Thank you for reviewing our work. We have carefully responded to your comments and questions in our rebuttal. Since the deadline for the author-reviewer discussion is fast approaching, could you please take a moment to review our responses? Any additional comments or feedback would be appreciated.
> >
> > Thank you

---

> > > ### Comment · Reviewer_rKCA · 2024-12-03
> > >
> > > Dear authors,
> > >
> > > I warmly thank you for your detailed response and revision. I did not go through all the details of the revision, but I can see that many of my comments on clarity are reflected. When I am certain that clarity issues are fully resolved, I think I will discuss with other reviewers about the significance of contributions, as this point is shared by me and some other reviewers.
> > >
> > > **Question.** Could you summarize the core contributions of your code base? Are they (1) the ability to make a "comprehensive comparison" and (2) implementation of "failure modes of CSM"?

---

> > > > ### Author Response · Authors · 2024-12-04
> > > >
> > > > Dear reviewer,
> > > >
> > > > Thank you for reading and acknowledging our rebuttal efforts. We are happy to have suitably addressed your initial concerns. We summarize our codebase below.
> > > >
> > > > At the model level, we have
> > > > - __a)__ the inclusion of higher-order Taylor expansion with both forward and reverse-mode AD
> > > > - __b)__ The ability to parameterize the context vector as a neural network ($i$CSM)
> > > >
> > > > These innovations are compatible with several deterministic and stochastic model-agnostic training algorithms:
> > > > - __i)__ bi-level optimization [1]
> > > > - __ii)__ ordinary alternating minimization [2]
> > > > - __iii)__ proximal alternating minimization [2]
> > > > - __iv)__ proximal alternating linearized minimization [3]
> > > >
> > > > Our library is highly modular, with all datasets exhibiting the same interface across data modalities (physical systems, images, etc.). Essentially, new datasets can be brought in with ease.
> > > >
> > > > Our library makes the combination of all these points seamless, thus enabling in this paper, (1) the comprehensive comparison and (2) the investigation of failures modes of CSM like the reviewer pointed out. For instance, the combination of points a) and i) above gave us the powerful FlashCAVIA as discussed in our paper.
> > > >
> > > > We note that a lot more is possible with our open-source code, and we are confident the community will contribute architectures, algorithms, and more, culminating in a unified repository for _gradient-based_ meta-learning experimentation. To the best of our knowledge, this is the first initiative to attempt coalescing the fast and growing body of research in this field.
> > > >
> > > >
> > > > ### References:
> > > >
> > > > [1] Zintgraf et al., Fast context adaptation via meta-learning, ICML, 2019.
> > > >
> > > > [2] Nzoyem et al, Neural Context Flows for Meta-Learning of Dynamical Systems, arXiv, 2024.
> > > >
> > > > [3] Driggs. et al. A stochastic proximal alternating minimization for nonsmooth and nonconvex optimization, SIAM, 2021.

---

### Official Review · Reviewer_ACqF · 2024-10-27

**Soundness:** 3
**Presentation:** 2
**Contribution:** 2
**Rating:** 3
**Confidence:** 3

**Summary:**

The paper extends Neural Context Flows (NCFs) [1], a recent method also in submission to this same conference, designed for meta-learning in dynamical systems applications. In particular, this work: 1) Extends a component of NCF (named iCSM) by applying it on the parameters of small networks (rather than finite vectors as in the original paper) 2) Proposes a new stochastic training strategy where the meta-learning tasks to optimize are randomly subsampled at each iteration. 3) Provides a new updated implementation of the CAVIA [2] algorithm and shows its compatibility with iCSM.

[1] Nzoyem, Roussel Desmond, David AW Barton, and Tom Deakin. "Neural Context Flows for Learning Generalizable Dynamical Systems." arXiv preprint arXiv:2405.02154 (2024).

[2] Zintgraf, Luisa M., et al. "Fast Context Adaptation via Meta-Learning." arXiv preprint arXiv:1810.03642 (2018).

**Strengths:**

1. There are several different orthogonal incremental contributions (e.g., iCSM, StochasticNCF, FlashCAVIA) with different purposes, which seem valuable and natural extensions to the NCF paper.

2. I found some of the qualitative analyses of the smoothing behavior, from introducing CSM to FlashCAVIA, insightful in obtaining an intuitive understanding of the methodology.

3. Overall, the writing is relatively clear, with most of the related background for understanding CSM and NCF introduced early on.

4. While I did not carefully check the code, the paper and results seem fully reproducible.

**Weaknesses:**

Main:

1. On its own, each contribution seems relatively minor e.g., to my understanding, stochasticNCF simply corresponds to randomly subsampling a set of environments during backpropagation of the meta-gradient. Similarly, FlashCAVIA is simply a port of CAVIA onto Jax, with minor changes to integrate it with CSM. While I did not carefully check the code, there do not seem to be new CUDA kernels introduced, as the name would seem to suggest. Thus, I am unsure whether these contributions are of enough novelty to stand as a full paper.

2. Given the incremental nature of this work over a still-not-established, recent prior method, I am not convinced of its relevance for the broader ICLR community.

3. The majority of the experimental evaluation is dedicated to toy domains (e.g., sine curve fitting/parametric ODEs). In contrast, the closely related CAVIA,  while being from 2018, still provided results on canonical benchmarks such as mini-imagenet and relatively complex Mujoco RL environments. While surely this paper's results can be insightful, they did not convince of the general applicability and potential impact of the methodology.

Minor:

1. Overuse of acronyms/abbreviations (e.g., InD, OoD, CSM, CNP, GBML, NCFs,...) can make the text hard to follow.

**Questions:**

Please, I would appreciate it if the authors could address the criticism raised above and, potentially, correct any mistakes in my current understanding of this work.

---

> ### Author Response · Authors · 2024-11-25
>
> Thank you for a great summary of our contribution, for praising the various incremental contributions we've made, each with a valuable purpose. It was our pleasure to introduce the methods in a way that was not only new and intuitive, but also unified across contextual meta learning families of methods. We are happy you found our qualitative results insightful, and our writing clear. The weaknesses you pointed out were addressed below.
>
> ---
>
> ### W1. Enough novelty ?
> We agree with the understanding the author has regarding our individual contributions, particularly the absence of CUDA kernels, although we wouldn't necessarily describe __all__ of them as "minor" or "simple" (particularly the inclusion of CSM into FlashCAVIA).
>
> Taken together, we argue that these individual contributions are indispensable for performing a __comprehensive comparison__ of several contextual meta-learning approaches. Beyond the changes, we provide a __codebase__ to combine various strategies together for faster research, and we argue that our conclusions that delineate clear __failure modes of CSM__ will be valuable to this community.
>
> Finally, we understand that naming our modification FlashCAVIA could mislead readers into relating it to FlashAttention [1]. We can only hope that won't be the case, and that readers will evaluate the method based on the updated description we provided in __Section 2.3__ of the new manuscript.
>
> [1] Dao et al. FlashAttention: Fast and Memory-Efficient Exact Attention with IO-Awareness, arXiv https://arxiv.org/abs/2205.14135, 2022
>
> ---
>
> ### W2. Still not established method ?
> We agree that while one work is fully established [1], the other work our __comparison__ builds upon is yet to not established [2]. We remark that, we've "simply" extracted one easy component of NCF [2] and applied it to other frameworks. We note that using __Taylor expansion for physical systems was introduced at ICLR last year [3], albeit limited to affine physical systems__. Our point is that regardless of how-well established the works we are comparing are, the underlying idea of Contextual Self-Modulation (CSM) is an interesting one that merits full investigation.
>
> Also, we think the value of our work is not tied to the success of [2] because if we remove NCF from this work, then some conclusions we've would remain true and valuable (e.g. FlashCAVIA + CSM is limited).
>
> [1] Zintgraf et al. Fast Context Adaptation via Meta-Learning, ICML 2019
>
> [2] Nzoyem et al. "Neural Context Flows for Learning Generalizable Dynamical Systems" ICLR24 Workshop on AI4DifferentialEquations, 2024.
>
> [3] Blanke et al., Interpretable Meta-Learning of Physical Systems, ICLR, 2024.
>
> ---
>
> ### W3. Toy problems
> We have made great efforts to provide problems in as many modalities as possible. Meta-Reinforcement Learning (and Mujoco RL) is not one of the topics we investigate in this work, although we agree it would be interesting to see how these methods perform in that setting.
>
> On the canonical side, we've included the CelebA dataset, on which we've drawn valuable conclusions. We believe the experiments we conducted, particularly on sine regression and image completion, are __enough to underscore several limitations of CSM__, which we believe the ICLR community would want to know about.
>
> ---
>
> ### Minor Weakness
> We thank you for pointing this out. We agree that the overuse of acronyms makes our texts hard to read. This is inevitable due to the comparative nature of our work. That said, we have added __Table 4__ in the appendix in an attempt to mitigate this. There, we define the various acronyms and their original references if possible.
>
> ---
>
> Your feedback has helped us improve our work, and we thank you very much for that. We are happy to address any other concerns you may have.

---

> > ### Comment · Reviewer_ACqF · 2024-11-26
> > **Response to Authors**
> >
> > I thank the authors for their response. I understand the three improvements proposed might be very valuable if proven effective, and I see utility in sharing code. However, given the current incremental nature of the work, without a thorough experiment section that investigates relevant problems beyond toy domains (which currently focuses on lower-scale problems than the 2018 paper CAVIA, which the authors claim this method is building upon) I fail to be convinced about its relevance for the broader ICLR community.
> >
> > Thus, I still do not think this work is ready in its current state and I would encourage the author to improve their work by focusing especially on the empirical aspect to provide convincing evidence about its effectiveness and scalability.

---

> > > ### Author Response · Authors · 2024-11-27
> > >
> > > Dear reviewer, we thank you for acknowledging our rebuttal efforts and for recognizing our improvements, along with the utility for sharing our code.

---

### Official Review · Reviewer_pMHW · 2024-11-02

**Soundness:** 2
**Presentation:** 2
**Contribution:** 2
**Rating:** 5
**Confidence:** 4

**Summary:**

This paper introduces a novel approach to meta-learning in high-data regimes by combining contextual self-modulation with CAVIA, an efficient optimization-based meta-learning framework. The main goal of this methodology is to expand the use of context information to infinite-dimensional spaces, addressing challenges such as handling a high volume of predictions when incorporating context. Additionally, the method employs a high-order Taylor expansion to effectively leverage context information for enhanced predictions.

To validate their approach, the authors conducted experiments on optimal control and forced pendulum tasks, demonstrating its applicability within scientific machine learning contexts. The results indicate that this model offers a more generalizable and efficient way to incorporate contextual information across low, medium, and high-data regimes, outperforming existing methods in both interpolation and extrapolation.

**Strengths:**

- This paper provides experimental validation showing that the proposed method maintains stability across a range of hyperparameter values, such as the order of Taylor expansion and the number of training steps. The results demonstrate that, even with variations in these newly introduced hyper-parameters, the method consistently outperforms previous algorithms, highlighting its robustness in hyper-parameter selection in curve-fitting experiments.

- This study explores the application of gradient-based meta-learning in the context of image completion. Specifically, family of gradient-based meta learning has traditionally seen limited success, on the other hand, this paper combining well-suited context self modulation structures illustrates that optimization-based meta-learning can indeed be effective for image completion, even in high-number prediction settings, thus broadening the potential applications of gradient-based meta learning approaches.

**Weaknesses:**

- This paper lacks a clearly defined contribution compared to existing models. This makes its core message difficult to understand. While the authors emphasize the performance of Context Self Modulation, high-order Taylor expansion, and FlashCAVIA or NCF as the backbone, the proposed method appears simple combination of these techniques on selected tasks. To convince the proposed method’s efficacy, I believe that a clearer illustration of the limitations of existing models and the way this approach effectively addresses would be necessary compared to existing methods. However, a majority part of the paper is to describing components to enhance the impression of this paper rather than substantiating significant performance improvements.

- While the paper emphasizes the potential of Context Self Modulation, its application is restricted to models like FlashCAVIA and NCF. Although the authors suggest broad applicability based on a variety of experiments, the focus remains on algorithms tailored to these specific frameworks, which may exaggerate the method’s generalizability.

- The necessity of high-order Taylor expansion for effectively leveraging context information is still ambiguous. The authors propose that this component is crucial, but Figure 1 does not show a substantial performance impact. This raises the question of whether a simpler, linear use of context information could be equally effective. Without compelling evidence for the necessity of high-order Taylor expansion, the paper does not convincingly establish why this feature is essential to the proposed method.

- The paper would benefit from the schematic figures to provide a clearer overview of the methodology. Additionally, numerous notational errors and inconsistencies impede understanding of the finer details. For instance, symbols such as $k$ and $K$ are used interchangeably despite having different meanings, and there is no clear explanation distinguishing $N$ from $K$. These issues with notation are unstructured and make it challenging for readers to follow the paper’s technical details accurately.

**Questions:**

- This paper requires additional parameters for Context Self Modulation compared to models like Neural Processes or MAML, yet does not address this increase anywhere in the discussion. To address the limitations of gradient-based methods, Context Information is utilized, but a clear comparison of this approach against existing methods is necessary. Specifically, it would be helpful to see how the number of Context Self Modulation parameters or the nature of the task impacts performance relative to established models. Especially for curve fitting experiments including forced pendulum and optimal control.

- Secondly, Figure 3 reveals significant underfitting, raising questions about the actual impact of Context Self Modulation. Given the observed underfitting in these experimental results, it remains unclear how effectively Context Self Modulation contributes to model performance. A closer examination of whether Context Self Modulation is indeed causing this underfitting would clarify its role in the training process.

**Details Of Ethics Concerns:**

No concern.

---

> ### Author Response · Authors · 2024-11-25
>
> Dear reviewer, we thank you for reading our work for providing valuable feedback. The fact that you found our experiment validation consequential is important to us. We are also please that you found our exploration of the limitations of gradient-based meta-learnig elucidating. Your concerns were addressed below.
>
> --
>
> ### W1. Weakly-defined contribution ?
> We agree with the reviewer that our core message wasn't efficiently emphasized. To clarify, our overall message is "CSM is effective on dynamical modelling tasks with smooth functions to learn, but it breaks when those assumptions are not met." The corresponding experiments and combination of various techniques introduced are at their core meant to validate this hypothesis. We've adjusted our wording in the abstract (lines __30-31__) to make clear what this core message is.
>
> Our __Table 1__ is meant as an illustration of the general weaknesses of each of these methods. However, due to space limitations, we couldn't discuss those limitations in the main text. Instead, we discussed those limitations in the __Appendix A__. We add that although it improves results on some tasks, CSM introduces an extra level of complexity into CAVIA [1]. This is contrary to our other contribution iCSM and StochasticNCF which are designed to learn infinite-dimensional functions and improve memory-scalability, respectively. We've clarified this in our contributions, around __line 152__.
>
> Beyond performance improvements, we focused on the MSE and the modelling capacity. A method like StochasticNCF was developed because the original NCF couldn't run on our hardware. This said, our exposition of StochasticNCF focuses on the accuracy of the gradient estimation, rather than size of $|B|$. We agree that this poses a disconnect with our performance claims when describing FlashCAVIA. To that end, we've reworded our description of FlashCAVIA to make it clear that __its most important addition is CSM__. We've added the term "Some" before "key" to indicate that the listed terms are not the main ingredients of the approach.
>
> ---
>
> ### W2. Can CSM be used outside NCF and CAVIA ?
> Based on its conceptual simplicity (i.e. Taylor expansion), we believe CSM indeed can be combined to various other methods. However, the implementation might be tricky depending on the base contextual meta-learning method, like we point out with FlashCAVIA. Considering the potential gains, this work was intended to investigate whether such implementations should be attempted, and we (informally) conclude that "it depends on the problem".
>
> We've removed the term "broad" in our introduction. We agree that its inclusion would suggest that CSM is broadly applicable to other problem, which isn't something we believe nor show.
>
> ---
>
> ### W3. Why high-order Taylor expansion ?
> We agree with the reviewer that the efficacy of higher-order Taylor expansion is not demonstrated in this work. Indeed, our work indicates the benefits of CSM with order 1, but not with higher-orders. Our other conclusions are equally conservative. We've redone our wording, which wouldn't be appropriate if we implied that higher-orders were crucial. We've made sure the term "crucial" is only used when there's evidence to suggest so.
>
> As originally demonstrated in [1], higher-order Taylor expansions are not essential for the problem. We've simply showed how one could implement them beyond $k>2$ efficiently with Taylor-Mode AD, and our conclusion is that despite Table 3, it is indeed not that important (__line 408__).
>
> Finally, our Results Synthesis 4.1 now makes clear the limitation of all these approaches we added.
>
> [1] Nzoyem et al. "Neural Context Flows for Learning Generalizable Dynamical Systems" ICLR24 Workshop on AI4DifferentialEquations, 2024.
>
> ---
> ### W4. Schematic picture to provide an overview
> We thank you for suggesting this. We have added a schematic illustrating the CSM mechanism in the new __Figure 1__ of the revised PDF. Concerning the notations, they are generally defined in the problem setting (Section 1.1) and further clarified depending on the problem. We apologise for the typo in Figure 3's caption, where we accidentally used $k$ rather than $K$.
>
> ---

---

> > ### Author Response · Authors · 2024-11-30
> >
> > Dear Reviewer pMHW,
> >
> > We appreciate your time and effort in reviewing our work. In response to your comments and queries, we have provided detailed answers in our rebuttal. With the discussion deadline drawing near, we kindly invite you to take a moment to review our latest updates.
> >
> > We are grateful for any further input or suggestions you might have.
> >
> > Thank you,

---

> > > ### Comment · Reviewer_pMHW · 2024-12-02
> > > **Response to authors rebuttal**
> > >
> > > The author’s rebuttal and updated manuscript provide detailed information and offer insights into the author’s intentions. However, the current version of the paper requires significant revisions to meet the standards of prestigious AI conferences.
> > >
> > > One of the unresolved issues in the manuscript is the lack of a clear and consistent message. The paper approaches the problems identified in existing research by treating each issue as an isolated challenge. This approach results in a lack of consistency in the research problem and the proposed method.
> > >
> > > Especially, the introduction of the high-order Taylor expansion, which is largely unrelated to the performance of the proposed method, is another notable concern. While the authors raise issues regarding its use, this element remains a methodological choice rather than addressing broader, fundamental challenges in meta-learning. The paper fails to identify or resolve the core issues typically associated with meta-learning frameworks.
> > >
> > > To be considered for acceptance in a prestigious AI conference, the paper must emphasize the broader challenges in meta-learning and present a clear, consistent statement. It should move beyond simply demonstrating performance improvements over existing methods and articulate how the proposed approach addresses identified challenges and contributes to the field.
> > >
> > > In its current form, the manuscript’s section-by-section focus on incremental improvements over prior works results in a scattered structure that detracts from its overall impact. In conclusion, I lean toward maintaining the current score.

---

> ### Author Response · Authors · 2024-11-25
>
> ### Q1. Comparison against established methods
>
> Again, we agree with the reviewer that it would be helpful to see how the number of Context Self Modulation parameters or the nature of the task impacts performance relative to established non-contextual models. This is something the two pieces of work we leverage establish effectively: CAVIA [1] is compared to MAML [2] and CNPs [3] across tasks and parameter count; NCFs does something similar with CoDA [4] and CAVIA [1]. We make ample references to those works and build on their results throughout. This is to say that the scope of our work is limited to contextual meta-learning [1,4,5], and we point the readers to the original works for further comparison outside non-contextual meta learning.
>
> Concerning varying context sizes, we point out in the __Table 2__ caption that the results are averaged across context sixes $d_\xi$. The same goes for __Figure 5__. Concerning the Optimal Control task, the number of context parameters was implicitly dictated by the dimensionality of the problem $d_{\xi} =2 $ (we found that using higher-dimensional contexts resulted in extremely poor performance). For the forced pendulum, we used an uncharacteristically large context vector $d_{\xi}=256$ for CSM so that all baselines could converge (see __footnote 4__). Some of these details were further clarified in __Appendix B.4__.
>
> Finally, the models in NCF [5] are the same ones used for CAVIA [1]. The differences are only in their optimization procedures (See __Algorithms 1 and 2__). We believe this makes our comparison of these two (the main focus of this work) fair and useful.
>
> [1] Zintgraf et al. Fast Context Adaptation via Meta-Learning, ICML 2019
> [2] Finn et al. Model-Agnostic Meta-Learning for Fast Adaptation of Deep Networks, ICML 2017
> [3] Garnelo et al. Conditional Neural Processes, ICML 2018
> [4] Kirchmeyer et al., Generalizing to New Physical Systems via Context-Informed Dynamics Model, ICML, 2022
> [5] Nzoyem et al. "Neural Context Flows for Learning Generalizable Dynamical Systems" ICLR24 Workshop on AI4DifferentialEquations, 2024.
>
> ---
>
> ### Q2. Underfitting with CSM
> Indeed, we agree, and we mention that CSM is the cause of the underfitting in Figure 3. The higher the order $k$, the worse the prediction gets. The explanation is associated to this effect was clarified in the surrounding text (__line 509__), and was further taken into account in delivering our final conclusion on the limitations of CSM in the revised PDF.
>
> ---
>
> We wish to sincerely thank for all these great remarks. It is clear from the revised PDF that they have made our presentation much clearer, and its contribution better emphasised.

---

> ### Author Response · Authors · 2024-12-04
>
> Dear reviewer,
>
> We thank you for acknowledging our rebuttal and for providing further feedback.
>
> Our hope with this paper is to show that three significant problems can be tackled in a unified manner. The three requirements we state in the introduction (Task Modality, Task Dimensionality, and Data Regime) each identify a problem to solve.
>
> While we present these issues separately, our modular _codebase_ allows users to work on them seamlessly, thus enabling deep experimentation. Interestingly, the removal of Taylor-expansion is easily performed by specifying its order as `0`, which still leaves us a valuable tool for comparing several optimization algorithms. We are confident the ICLR community will benefit from it.
>
> We did not emphasize the usefulness of our codebase in our original paper, but we believe it can easily be addressed in the final version of the paper.
>
> Once again, thank you for your feedback.
>
> The Authors

---

### Official Review · Reviewer_KX5K · 2024-11-03

**Soundness:** 2
**Presentation:** 2
**Contribution:** 2
**Rating:** 5
**Confidence:** 3

**Summary:**

In this work, the authors investigate model based meta-learning approaches that involve learning a dataset-specific context for learning function changes across datasets, which they refer to as contextual self modulation (CSM). They specifically introduce an infinite-dimensional, stochastic meta-learning approach that extends recent work and apply the method to different settings including image completion, control and dynamical system reconstruction.

**Strengths:**

Context-based meta learning has become a popular approach for training models that can adapt to new data in few-shot settings. There are limited studies investigating the strengths and weaknesses of these methods – making this an overall well motivated and interesting problem to address. The authors report the performance of their approach under diverse experimental settings, demonstrating the wide applicability of CSM to different domains. That being said, I have various concerns that I highlight below.

**Weaknesses:**

Presentation
* In the introduction, the authors motivate the paper as a systematic comparison of contextual meta-learning approaches but only focus on a recently proposed method Neural Context Flow (NCF) and Cavia which seems a little misleading.
In the Introduction R2 (line 064), I find the authors’ introduction of task dimensionality a little confusing. At the beginning, it seems like the authors are referring to function spaces like vector fields for which several approaches have been proposed but then discuss a non-stationary variable as an example. I could be misunderstanding something, but it would be useful to further clarify this point and how exactly the iCSM addresses this.
* The main contributions – the infinite-dimensional CSM and stochastic NCF could to be better motivated and further expanded because it is a little unclear. Specifically, I don’t fully understand the motivation for iCSM and how it is different from approaches that use context-conditioned hypernetworks, like CoDA. In lines 192-194, the authors claim that “This extension allows our model $f_{\theta}$ to adapt to a broader range of changes and potentially leverage Lie group symmetries for dynamical systems and translation equivariance for images” – I don’t see why this would be the case.
* While FlashCavia offers strong improvements over Cavia, I don’t see how it relates to the proposed approach, unless I’m misunderstanding something.

Experiments
* The empirical results are highly variable across experiments. While it is not important to have the best performance across the board, the lack of consistent trends makes it hard to evaluate the proposed approach. For instance, FlashCAVIA has the best performance on Sine Regression, in the Image Completion task, the performance does not co-vary with K for either approach considered.
* Limited baselines are considered, and they are not evaluated consistently across experiments. If there are optimization issues because of the higher order gradients in Cavia, it would still be interesting to consider how the specific parametrization they proposed compares to the NCF/iNCF. In general, a more thorough related works section and additional baseline comparisons would be helpful.

**Questions:**

* Why do the authors use NCF* (line 428) in the image completion task?
* From what I understand, CoDA relies on low-dimensional adaptation of weights. Why do the authors use a context size of 256? In the original paper, the method is only used with a context dimensionality of 1 or 2. It would be good to additionally include the case where the context is low-dimensional.
* Did the authors compare their approach to CAMEL (Blanke & LeLarge, 2024)? It also seems like an important baseline to consider, specifically, for measuring performance with higher-order Taylor expansion terms.

---

> ### Author Response · Authors · 2024-11-25
>
> Thank you for reading and for reviewing our work. Given the popularity of context-based meta learning, we too believe this is an important field, as various existing studies can attest. Thank you for finding our motivation for the problem adequate, and for noticing the wide range of our experiments that bring CSM into several domains. We've addressed your concerns below.
>
> ---
>
> ### W1. Focus on NCF [1] and CAVIA [2]
> We focus on NCF and CAVIA because those are two representatives of two closely-related families of contextual meta-learning: __bi-level__ optimisation vs alternating minimisation. The other family, NP offers a completely different paradigm, and its inclusion in Table 1 hopefully helps readers understand benefits of the gradient based paradigm. We agree that our sentence is misleading, and we've rephrased it to "comprehensive comparison" (__line 051__).
>
> What do we mean by __task-dimensionality__ ? We mean adaptation to functions, rather than stationary variables. We realise that we referred to learning a vector field and leaning its parameter dependence as the same thing, which might be confusing. We've modified the section iCSM to indicate that it models its context vector as the flattened weights of a neural network (This is also made clear in our new __Figure 1__). By doing so, we are learning the weights of another network (albeit smaller than the base network to which this context is fed).
>
> [1] Nzoyem et al. "Neural Context Flows for Learning Generalizable Dynamical Systems" ICLR24 Workshop on AI4DifferentialEquations, 2024.
>
> [2] Zintgraf, Luisa M., et al. "Fast Context Adaptation via Meta-Learning" ICML 2018.
>
> ---
>
> ### W2. The motivation for iCSM
>
> The main motivation behind iCSM is addressed in W1 above. We agree that our passage on Infinite-dimensions did not do enough to clarify how iCSM works. We have removed the sentence mentioned by the reviewer, and replaced it with explanations of the workings of the iCSM approach (__line 210__).
>
> ---
>
> ### W3. Relation between FlashCAVIA and CAVIA
>
> Like the reviewer noticed, FlashCAVIA offers performance benefits over CAVIA. But those "key" benefits were originally stated asif they were all implementation-based.
>
> We have now updated our manuscript to show that FlashCAVIA in __section 2.3__ incorporates the contextual-self-modulation (CSM) mehanism presented in [1], which is why it is suited for comparison in this work.
>
> [1] Nzoyem et al. "Neural Context Flows for Learning Generalizable Dynamical Systems" ICLR24 Workshop on AI4DifferentialEquations, 2024.
>
> ---
>
> ### W4. Trends in the results
> We agree that there is no best performance across the board. The CSM approach introduced in [1] and extended by us is indeed powerful, but still limited. We particularly agree with the reviewer regarding the hyperparameter $k$ which causes underfitting in __Figure 3__, but no noticeable improvements in __Figure 1__. We've rewritten our first paragraph of __4.1 Results Synthesis__ to make these conclusions clearer.
>
> ---
>
> ### W5. More thorough related work section and baselines
> Indeed, we agree that a more thorough related work could be helpful. We did our best to include as many families of methods as possible while limiting ourselves to the most relevant methods in each. Otherwise, this endeavour would have been unmanageable.
>
> Specifically, on each problem, we considered different baselines always ensuring we were fair. We agree, consistency across problems is lacking, but we argue that this is hard to mitigated against, due to the inherent variations is modalities involved in these tasks. All in all, we suggest that our results be interpreted on a case-by-case basis. We've additionally updated our Results Synthesis to extract general conclusions and save time for readers of our work.
>
> Finally, we were limited in our Related Works section due to space concerns. Because of that, we were forced to present the Main Related Works in the __Appendix A__. We agree with the reviewer that this isn't ideal, but this felt like the best alternative at our disposal.
>
> ---
>
> ### Q1. Why do we use NCF* ?
> This approach is simply faster than NCF. In fact, it is twice as fast since the neural network's weights and contexts can be differentiated against and updated in one go, instead of separately as with __alternating__ minimisation. We've made this clear around that same __line 465__.
>
> ---
>
> ### Q2. Why not use a low-dimensioanl context for CoDA
> We agree that it would have been good to use CoDA in the setting it was designed. But doing so resulted in early stagnation of the loss curve. Intuitively, we believe this is because the parameter we are trying to model with the forced pendulum is infinitely-dimensional, whereas CoDA was designed for finite-dimensional changes. Increasing this value allowed us to observe the curve in __Figure 2__, against which we could perform a comparison. We pointed out these difficulties in the __footnote 4__ of the original PDF, now page 8.
>
> ---

---

> ### Author Response · Authors · 2024-11-25
>
> ### Q3. Comparison with CAMEL
> Thank you once more for pointing out CAMEL [4]. We completely agree that this work is relevant to ours, and particularly to CoDA [5], like we pointed out in the Extended Related Work section (Appendix A). However, CAMEL is focused on interpretability and aimed at physically-affine physical systems, whereas none of our problems are affine. This motivated our focus on improving only CoDA as a baseline and making it work for our hard cases.
>
> [4] Blanke et al., Interpretable Meta-Learning of Physical Systems, ICLR, 2024.
>
> [5] Kirchmeyer et al., Generalizing to New Physical Systems via Context-Informed Dynamics Model, ICML, 2022.
>
> ---
>
> Once more, thank you. Your questions have helped us clarify the paper, particularly the iCSM mechanism.

---

> > ### Author Response · Authors · 2024-11-30
> >
> > Dear Reviewer KX5K,
> >
> > Thank you for your thoughtful evaluation of our work. We’ve taken care to address your questions and feedback in our rebuttal. As the discussion deadline approaches, we’d greatly appreciate it if you could review our responses at your convenience.
> >
> > If you have any further thoughts or comments, we’d be happy to address them promptly.
> >
> > Thank you,
> > The Authors

---

> > > ### Comment · Reviewer_KX5K · 2024-12-01
> > >
> > > I thank the authors for responding to my concerns and making edits to the manuscript to improve the overall clarity. There are some interesting contributions to make the methods implemented computationally efficient and I have raised my score based on the authors' response. However, I still have concerns about the lack of trends in the empirical results. I encourage the authors to further investigate the experimental results, and make the narrative more focused.

---

> > > > ### Author Response · Authors · 2024-12-02
> > > >
> > > > Dear reviewer,
> > > >
> > > > We thank you for acknowledging our rebuttal efforts and for adjusting your score accordingly. We will make sure our final version emphasizes that overarching trends in empirical results will be investigated in future work, along with what they might entail.
> > > >
> > > > Thank you
> > > >
> > > > The Authors

---

### Author Response · Authors · 2024-11-25

Dear reviewers R1(KX5K), R2(pMHW), R3(ACqF), and R4(rKCA), we are grateful for your feedback on our work. Some of you found interesting the problem we are tacking in this field of context-based meta-learning (R1). The variety of our experiments, their validation, and robustness to hyperparameter selection is also a positive theme we noticed (R1, R2, R3), particularly emphasized on image completion and its qualitative analyses (R2, R3). We are also happy you appreciate the various scalable extensions we added in order to compare these methods on a wide spectrum of tasks.

Thank you for pointing out weaknesses in our paper. In addressing your concerns, we had to make several changes. These are all highlighted in red in the revised manuscript. One major change we performed is the addition of the schematic in __Figure 1__ which clearly illustrates the CSM mechanism, and how our $i$CSM contribution is different. This results in every Figure number of the original PDF incremented by 1. Due to space limitations, we've moved the results table on image completion to the Appendix (previously Table 4, now __Table 6__). This allows us to focus our discussion on the visibly reduced qualitative results with CSM. Finally, we've included __Table 4__ which defines most of the acronyms used in this work.

Your questions have undeniably helped strengthen the paper. We hope our responses are crisp and clear enough to be satisfactory, and we are happy to provide additional information to resolve any other concern you may have. Thank you.

---

### Meta-Review · Area_Chair_q1YY · 2024-12-23

**Metareview:**

Due to unanimous agreement among a few core criticisms, namely (a) Lack of novelty/contribution (b) Presentation/Clarity (c) Impact on the ICLR community, this leads, unfortunately, to an unambiguous decision against publication at ICLR.

**Additional Comments On Reviewer Discussion:**

This submission saw a very positive discussion between reviewers and authors, with all reviewers engaging with the authors' rebuttal.

---

### Decision · Program_Chairs · 2025-01-22

Reject